# Offline RL via Feature-Occupancy Gradient Ascent

## Abstract

We study offline Reinforcement Learning in large infinite-horizon discounted Markov Decision Processes (MDPs) when the reward and transition models are linearly realizable under a known feature map. Starting from the classic linear-program formulation of the optimal control problem in MDPs, we develop a new algorithm that performs a form of gradient ascent in the space of feature occupancies, defined as the expected feature vectors that can potentially be generated by executing policies in the environment. We show that the resulting simple algorithm satisfies strong computational and sample complexity guarantees, achieved under the least restrictive data coverage assumptions known in the literature. In particular, we show that the sample complexity of our method scales optimally with the desired accuracy level and depends on a weak notion of coverage that only requires the empirical feature covariance matrix to cover a single direction in the feature space (as opposed to covering a full subspace). Additionally, our method is easy to implement and requires no prior knowledge of the coverage ratio (or even an upper bound on it), which altogether make it the strongest known algorithm for this setting to date.

## 1 Introduction

We study Offline Reinforcement Learning (ORL) in sequential decision making problems whereby a learner aims to find a near-optimal policy with sole access to a static dataset of interactions with the underlying environment [Levine et al., 2020]. This line of work is naturally relevant to real-world tasks for which learning an accurate simulator of the environment is potentially intractable or impossible, trial-and-error learning could have grave consequences, yet logged interaction data is readily available. For example, in a high-stake application such as autonomous driving, building a sufficiently accurate simulator for the vehicle and its environment would require modelling very complex systems, which can be intractable both statistically and computationally. At the same time, running experiments in the real world could endanger the lives of other road users or result in damages to the vehicle. Yet, with the advent of tools for efficient sensory-data collection and processing, large volumes of logged data from human drivers are readily available.

An efficient ORL method is one which finds a near-optimal policy after a tractable number of elementary computations and samples from the dataset. It is well-known in this setting that the quality of the solution has to heavily depend on the quality of the data, and in particular one cannot hope to find a near-optimal policy if the data covers the space of states and actions poorly. To formalize this intuition, many notions of data coverage have been proposed in the offline RL literature, ranging from a very restrictive uniform coverage assumption that requires the data-generating policy to cover the entire state-action space [Munos and Szepesvári, 2008] to a variety of partial coverage conditions whereby this exploratory condition is only required for state-action pairs that are of interest to the optimal policy [Liu et al., 2020, Rashidinejad et al., 2021, Uehara and Sun, 2021, Zhan et al., 2022, Rashidinejad et al., 2022, Li et al., 2024]. In the present work, we study the setting of linear *Markov*

*Decision Processes* (MDPs) [Jin et al., 2020, Yang and Wang, 2019] where the reward and transition matrix admit a low rank structure in terms of a known feature map, and data-coverage assumptions can be defined in the space of features. As shown by [Zanette et al., 2021], in this setting it is possible to obtain strong guarantees if the offline data is well-aligned with the expectation of the feature vector generated by the optimal policy (as opposed to requiring alignment with the entire distribution of features as required by other common offline RL methods [Jin et al., 2021, Xie et al., 2021, Uehara and Sun, 2021, Zhang et al., 2022]). In the present paper, we propose a simple and efficient algorithm that yields the best known sample complexity guarantees for this problem setting, all while only requiring the weakest known data-coverage assumptions of Zanette et al. [2021].

Our approach is based on the LP formulation of optimal control in infinite-horizon discounted MDPs due to Manne [1960], and more specifically on its low-dimensional saddle-point reparametrization for linear MDPs proposed by Gabbianelli et al. [2024] (which itself builds on earlier work by Neu and Okolo, 2023 and Bas-Serrano et al., 2021). Primal variables of this saddle-point objective correspond to expectations of feature vectors under the state-action distribution of each policy (called *feature occupancies*), and dual variables correspond to parameters of linear approximations of action-value functions. We design an algorithm based on the idea of optimizing the unconstrained primal function that is derived from the saddle-point objective by eliminating the dual variables via a classic dualization trick. More precisely, we design a sample-based estimator of the primal function and optimize it via a variant of gradient ascent in the space of feature occupancies.

This approach is to be contrasted with the method of Gabbianelli et al. [2024], which instead optimized the original saddle-point objective via stochastic primal-dual methods. Their algorithm interleaved a sequence of "policy improvement" steps with an inner loop performing "policy evaluation", which resulted in a suboptimal use of sample transitions due to the costly inner loop. This issue was addressed in the very recent work of Hong and Tewari [2024] who, instead of relying on stochastic optimization, built an estimator of the saddle-point objective and optimized it via a deterministic primal-dual method. Our approach is directly inspired by their idea of estimating the saddle-point objective, but our algorithm design is significantly simpler: instead of directly optimizing the primal function in terms of feature occupancies, Hong and Tewari [2024] relied on a sophisticated reparametrization of the primal variables, and used a computationally involved procedure to update the dual variables. Both of these steps required prior knowledge of a tight bound on the feature-coverage ratio of the optimal policy, which is typically not available in problems of practical interest. Such knowledge is not required by our algorithm, thanks to the incorporation of a recently proposed stabilization trick that we make use of in our algorithm [Jacobsen and Cutkosky, 2023, Neu and Okolo, 2024]. We provide a more detailed discussion of these closely related works in Section 5.

**Notation.** We use boldface lowercase letters $\boldsymbol{m}$ to denote vectors and and bold uppercase $\boldsymbol{M}$ for matrices. We define the Euclidean ball in $\mathbb{R}^d$ of radius $D$ by $\mathbb{B}_d(D) = \{\boldsymbol{x} \in \mathbb{R}^d | \, \|\boldsymbol{x}\|_2 \leq D\}$ and the $A$-simplex over a finite set $\mathcal{A}$ of cardinality $A$ as $\Delta_{\mathcal{A}} = \{p \in \mathbb{R}_+^A | \, \|p\|_1 = 1\}$.

## 2 Preliminaries

We consider infinite-horizon Discounted Markov Decision Processes (DMDPs) [Puterman, 1994] of the form $(\mathcal{X}, \mathcal{A}, \boldsymbol{r}, \boldsymbol{P}, \gamma)$ where $\mathcal{X}$ denotes a finite (yet large) set of $X$ states and $\mathcal{A}$ is a finite action space of cardinality $A = |\mathcal{A}|$. We refer to $\boldsymbol{r} \in [0, 1]^{XA}$ as the reward vector, $\boldsymbol{P} \in \mathbb{R}_+^{XA \times X}$ the transition matrix and $\gamma \in (0, 1)$ the discount factor. For a state-action pair $(x, a) \in \mathcal{X} \times \mathcal{A}$ we also use the notation $r(x, a) = \boldsymbol{r}[(x, a)]$ to denote the reward of taking action $a$ in state $x$ and $p(x'|x, a) = \boldsymbol{P}[(x, a), x']$ as the probability of ending up in state $x'$ afterwards.

The MDP models a sequential decision making process where an agent interacts with its environment as follows. For each step $k = 0, 1, 2, \cdots,$, the agent observes the current state $X_k$ of the environment and then goes on to select its action $A_k$. Based on this action in the current state, it receives a reward $r(X_k, A_k)$, transits to a new state $X_{k+1} \sim \boldsymbol{p}(\cdot|X_k, A_k)$ and the process continues. The objective of the agent is to find a decision-making rule that maximizes its total discounted reward when the initial state $X_0$ is sampled according to a fixed initial-state distribution distribution $\boldsymbol{\nu}_0 \in \Delta_{\mathcal{X}}$. Without loss of generality, we assume that the initial state is fixed almost surely as $X_0 = x_0$, and use $\boldsymbol{\nu}_0$ to refer to the corresponding delta distribution. It is known that this objective can be achieved by executing a *stationary stochastic policy* $\pi : \mathcal{X} \to \Delta_{\mathcal{A}}$, with $\pi(a|x)$ denoting the probability of the agent selecting action $A_k = a$ in state $X_k = x$ for all $k$. We will use $\Pi$ to denote the set of all such behavior rules

and will often simply call them *policies*. We define the normalized discounted return of each policy $\pi$ as

$$\rho\left(\pi\right) = \left(1 - \gamma\right) \mathbb{E}_{\boldsymbol{\nu}_0, \pi}\left[\sum_{k=0}^{\infty} \gamma^k r\left(x_k, a_k\right)\right],$$

where the role of the discount factor $\gamma \in (0, 1)$ is to emphasize the importance of earlier rewards, and the notation $\mathbb{E}_{\boldsymbol{\nu}_0, \pi}\left[\cdot\right]$ highlights that the initial state is sampled from $\nu_0$ and all actions are sampled according to the policy $\pi$. We will use $\pi^*$ to denote any policy that maximizes the return.

We will consider the offline RL setting where we are given access to a data set of $n$ sample transitions $\mathcal{D}_n = \{(X_i, A_i, R_i, X_i')\}_{i=1}^n$, where $X_i' \sim \boldsymbol{p}(\cdot|X_i, A_i)$ is sampled independently for each $i$ and $R_i = r(X_i, A_i)$. Otherwise, no assumption is made about the state-action pairs $(X_i, A_i)$, and in particular we do not require these to be generated by a fixed behavior policy or to be independent of each other.

For describing the approach we take towards solving this problem, we need to introduce some further standard notations. The value function and action-value function associated with policy $\pi$ are respectively defined as

$$v^\pi\left(x\right) = \mathbb{E}_{a \sim \pi(\cdot|x)}\left[q^\pi\left(x, a\right)\right], \quad q^\pi\left(x, a\right) = \mathbb{E}_\pi\left[\sum_{k=0}^{\infty} \gamma^k r\left(x_k, a_k\right) \middle| x_0 = x, a_0 = a\right],$$

and the state-occupancy and state-action-occupancy measures under $\pi$ as

$$\nu^\pi\left(x\right) = \sum_a \mu^\pi\left(x, a\right), \quad \mu^\pi\left(x, a\right) = \left(1 - \gamma\right) \mathbb{E}_{\boldsymbol{\nu}_0, \pi}\left[\sum_{k=0}^{\infty} \gamma^k \mathbb{I}_{\{x_k, a_k\}}\right].$$

The value functions and occupancy measures adhere to the following recursive equations, respectively termed the Bellman equation and Bellman flow condition [Bellman, 1966]:

$$\boldsymbol{q}^\pi = \boldsymbol{r} + \gamma \boldsymbol{P} \boldsymbol{v}^\pi, \qquad \boldsymbol{\mu}^\pi = \pi \circ \left[\left(1 - \gamma\right) \boldsymbol{\nu}_0 + \gamma \boldsymbol{P}^\mathsf{T} \boldsymbol{\mu}^\pi\right].$$

Here, the composition operation $\circ$ is defined so that for any policy $\pi$ and state distribution $\boldsymbol{\nu} \in \mathbb{R}^X$, we have $\left(\pi \circ \boldsymbol{\nu}\right)\left(x, a\right) = \pi\left(a|x\right) \nu\left(x\right)$. Notice that we can express the return of $\pi$ in terms of value functions and occupancy measures as $\rho\left(\pi\right) = \left(1 - \gamma\right) \langle \boldsymbol{\nu}_0, \boldsymbol{v}^\pi \rangle = \langle \boldsymbol{\mu}^\pi, \boldsymbol{r} \rangle$. On this note, for a given target accuracy $\varepsilon > 0$, we say policy $\pi$ is $\varepsilon$-optimal if it satisfies $\langle \boldsymbol{\mu}^{\pi^*} - \boldsymbol{\mu}^\pi, \boldsymbol{r} \rangle \leq \varepsilon$.

In the present work, we well make use of the *linear MDP* assumption due to Jin et al. [2020], Yang and Wang [2019], which is defined formally as follows:

**Definition 2.1** (Linear MDP). An MDP is called linear if both the transition and reward functions can be expressed as a linear function of a given feature map $\boldsymbol{\varphi} : \mathcal{X} \times \mathcal{A} \to \mathbb{R}^d$. That is, there exist $\boldsymbol{\psi} : \mathcal{X} \to \mathbb{R}^d$ and $\boldsymbol{\omega} \in \mathbb{R}^d$ such that, for every $x, x' \in \mathcal{X}$ and $a \in \mathcal{A}$:

$$r(x, a) = \langle \boldsymbol{\varphi}(x, a), \boldsymbol{\omega} \rangle, \quad p\left(x'|x, a\right) = \langle \boldsymbol{\varphi}(x, a), \boldsymbol{\psi}(x') \rangle.$$

We denote by $\boldsymbol{\Phi} \in \mathbb{R}^{|\mathcal{X} \times \mathcal{A}| \times d}$ the feature matrix with rows given by $\boldsymbol{\varphi}(x, a)^\mathsf{T}$ and $\boldsymbol{\Psi} \in \mathbb{R}^{d \times |\mathcal{X}|}$ as the weight matrix with columns $\boldsymbol{\psi}(x)$. Further, we will assume that $\|\boldsymbol{\omega}\|_2 \leq \sqrt{d}$, that $\|\boldsymbol{\Psi}\boldsymbol{v}\|_2 \leq B\sqrt{d}$ holds for all $\boldsymbol{v} \in [-B, B]$, and that all feature vectors satisfy $\|\boldsymbol{\varphi}(x, a)\|_2 \leq R$ for some $R \geq 1$.

An immediate consequence of this assumption is that the action-value function of any policy $\pi$ can be written as a linear function of the features as $\boldsymbol{q}^\pi = \boldsymbol{\Phi}\boldsymbol{\theta}^\pi$, with $\boldsymbol{\theta}^\pi = \boldsymbol{\omega} + \gamma \boldsymbol{\Psi}\boldsymbol{v}^\pi \in \mathbb{R}^d$. For the rest of the paper we explicitly assume that the feature matrix $\boldsymbol{\Phi}$ is full rank – which is enough to ensure uniqueness of $\boldsymbol{\theta}^\pi$. It is common to assume that the feature dimension $d \ll X$ such that the transition operator is low-rank. As common in this setting, we will suppose throughout the paper that the feature map $\boldsymbol{\Phi}$ is known.

Our algorithm design will be based on the linear programming formulation of MDPs, first proposed in a number of papers in the 1960's [Manne, 1960, de Ghellinck, 1960, d'Epenoux, 1963, Denardo, 1970]. This formulation frames the problem of finding an optimal control policy as the following pair of primal and dual linear programs:

$$
\begin{array}{ll}
\text{maximize} & \langle \boldsymbol{\mu}, \boldsymbol{r} \rangle \\
\text{subject to} & \boldsymbol{E}^\mathsf{T} \boldsymbol{\mu} = (1 - \gamma)\boldsymbol{\nu}_0 + \gamma \boldsymbol{P}^\mathsf{T} \boldsymbol{\mu} \quad (1) \\
& \boldsymbol{\mu} \geq 0,
\end{array}
\qquad
\begin{array}{ll}
\text{minimize} & (1 - \gamma)\langle \boldsymbol{\nu}_0, \boldsymbol{v} \rangle \\
\text{subject to} & \boldsymbol{E}\boldsymbol{v} \geq \boldsymbol{r} + \gamma \boldsymbol{P}\boldsymbol{v}. \quad (2)
\end{array}
$$

131 Here, the operator $\boldsymbol{E} \in \mathbb{R}^{XA \times X}$ is defined such that for each $x, a$ and vectors $\boldsymbol{\mu} \in \mathbb{R}^{XA}, \boldsymbol{v} \in \mathbb{R}^X$,

$$\left(\boldsymbol{E}^{\mathsf{T}} \boldsymbol{\mu}\right)(x) = \sum_{a \in \mathcal{A}} \boldsymbol{\mu}(x, a), \qquad \left(\boldsymbol{E} \boldsymbol{v}\right)(x, a) = \boldsymbol{v}(x).$$

132 It is known that the occupancy measure of an optimal policy $\boldsymbol{\mu}^{\pi^*}$ is an optimal solution of the primal
133 LP (1). In fact, the feasible set of the primal is precisely the space of valid state-action occupancy
134 measures that can be induced by stationary policies. Therefore, given any feasible solution $\boldsymbol{\mu}$, we can
135 extract the inducing policy as $\pi_\mu(a|x) = \mu(x, a) / \sum_{a'} \mu(x, a')$ when $\sum_a \mu(x, a) \neq 0$. Likewise,
136 the state value function of the optimal policy $\pi^*$ is an optimal solution to the dual LP. That said, since
137 the LP features $XA$ variables and constraints, it cannot be solved directly in large MDPs.

138 In view of the above limitations, we consider the following reduced version of the above intractable
139 LPs due to Gabbianelli et al. [2024] (see also Neu and Okolo, 2023, Bas-Serrano et al., 2021):

$$
\begin{array}{ll}
\text{maximize} & \langle \boldsymbol{\lambda}, \boldsymbol{\omega} \rangle \\
\text{subject to} & \boldsymbol{E}^{\mathsf{T}} \boldsymbol{\mu} = (1 - \gamma) \boldsymbol{\nu}_0 + \gamma \boldsymbol{\Psi}^{\mathsf{T}} \boldsymbol{\lambda} \\
& \boldsymbol{\lambda} = \boldsymbol{\Phi}^{\mathsf{T}} \boldsymbol{\mu} \\
& \boldsymbol{\mu} \geq 0,
\end{array}
\quad (3)
\qquad
\begin{array}{ll}
\text{minimize} & (1 - \gamma) \langle \boldsymbol{\nu}_0, \boldsymbol{v} \rangle \\
\text{subject to} & \boldsymbol{E} \boldsymbol{v} \geq \boldsymbol{\Phi} \boldsymbol{\theta} \\
& \boldsymbol{\theta} = \boldsymbol{\omega} + \gamma \boldsymbol{\Psi} \boldsymbol{v}.
\end{array}
\quad (4)
$$

141 In view of the second constraint of the primal LP (3), $\boldsymbol{\lambda}$ should be thought of as expectations of feature
142 vectors under occupancy measures, and we thus refer to them as *feature occupancy* vectors. Similarly,
143 the second constraint of the dual LP (4) suggests that $\boldsymbol{\theta}$ should be thought of as parameters of the
144 approximate action-value function $\boldsymbol{q}_{\boldsymbol{\theta}} = \boldsymbol{\Phi} \boldsymbol{\theta} = \boldsymbol{\Phi}(\boldsymbol{\omega} + \gamma \boldsymbol{\Psi} \boldsymbol{v}) = \boldsymbol{r} + \gamma \boldsymbol{P} \boldsymbol{v}$. We use $\boldsymbol{\lambda}^{\pi^*} = \boldsymbol{\Phi}^{\mathsf{T}} \boldsymbol{\mu}^{\pi^*}$ to
145 denote the feature occupancy associated with the optimal policy $\pi^*$ and $\boldsymbol{\theta}^{\pi^*}$ to denote the parameter-
146 vector of the optimal action-value function $\boldsymbol{q}^{\pi^*}$. The Lagrangian corresponding to the LPs is given as

$$
\begin{aligned}
\mathfrak{L}(\boldsymbol{\lambda}, \boldsymbol{\mu}; \boldsymbol{v}, \boldsymbol{\theta}) &= (1 - \gamma) \langle \boldsymbol{\nu}_0, \boldsymbol{v} \rangle + \langle \boldsymbol{\lambda}, \boldsymbol{\omega} + \gamma \boldsymbol{\Psi} \boldsymbol{v} - \boldsymbol{\theta} \rangle + \langle \boldsymbol{\mu}, \boldsymbol{\Phi} \boldsymbol{\theta} - \boldsymbol{E} \boldsymbol{v} \rangle \\
&= \langle \boldsymbol{\lambda}, \boldsymbol{\omega} \rangle + \langle \boldsymbol{v}, (1 - \gamma) \boldsymbol{\nu}_0 + \gamma \boldsymbol{\Psi}^{\mathsf{T}} \boldsymbol{\lambda} - \boldsymbol{E}^{\mathsf{T}} \boldsymbol{\mu} \rangle + \langle \boldsymbol{\theta}, \boldsymbol{\Phi}^{\mathsf{T}} \boldsymbol{\mu} - \boldsymbol{\lambda} \rangle.
\end{aligned}
\quad (5)
$$

147 It is easy to verify that by the linear MDP property, the feasible sets of the above LPs coincide
148 with those of the original LPs in an appropriate sense, and their optimal solutions correspond to
149 the optimal state-action occupancy measure and state-value function respectively (see Appendix A).

150 In order to further reduce the complexity of the LPs above, we introduce a policy $\pi$ and parametrize
151 the remaining high-dimensional variables $\boldsymbol{v}$ and $\boldsymbol{\mu}$ as

$$v_{\boldsymbol{\theta}, \pi}(s) = \sum_a \pi(a|s) \langle \boldsymbol{\theta}, \boldsymbol{\varphi}(x, a) \rangle, \qquad \mu_{\boldsymbol{\lambda}, \pi}(x, a) = \pi(a|x) \left[ (1 - \gamma) \nu_0(x) + \gamma \langle \boldsymbol{\psi}(x), \boldsymbol{\lambda} \rangle \right]. \quad (6)$$

152 Plugging this choice back into the Lagrangian, we obtain the objective

$$
\begin{aligned}
f(\boldsymbol{\lambda}, \pi; \boldsymbol{\theta}) &= \mathfrak{L}(\boldsymbol{\lambda}, \boldsymbol{\mu}_{\boldsymbol{\lambda}, \pi}; \boldsymbol{v}_{\boldsymbol{\theta}, \pi}, \boldsymbol{\theta}) \\
&= (1 - \gamma) \langle \boldsymbol{\nu}_0, \boldsymbol{v}_{\boldsymbol{\theta}, \pi} \rangle + \langle \boldsymbol{\lambda}, \boldsymbol{\omega} + \gamma \boldsymbol{\Psi} \boldsymbol{v}_{\boldsymbol{\theta}, \pi} - \boldsymbol{\theta} \rangle \\
&= \langle \boldsymbol{\lambda}, \boldsymbol{\omega} \rangle + \langle \boldsymbol{\theta}, \boldsymbol{\Phi}^{\mathsf{T}} \boldsymbol{\mu}_{\boldsymbol{\lambda}, \pi} - \boldsymbol{\lambda} \rangle.
\end{aligned}
\quad (7)
$$

153 It is easy to see that for any $\pi$ and $\boldsymbol{\lambda}^\pi = \boldsymbol{\Phi}^{\mathsf{T}} \boldsymbol{\mu}^\pi$, we have $f(\boldsymbol{\lambda}^\pi, \pi; \boldsymbol{\theta}) = \langle \boldsymbol{\mu}^\pi, \boldsymbol{r} \rangle$ for all $\boldsymbol{\theta} \in$
154 $\mathbb{R}^d$. Furthermore, whenever $\boldsymbol{\lambda} \neq \boldsymbol{\lambda}^\pi$ then the $\boldsymbol{\theta}$-player has a winning strategy that can force
155 $\min_{\boldsymbol{\theta}} f(\boldsymbol{\lambda}, \pi; \boldsymbol{\theta}) = -\infty$. This (informally) suggests that an optimal policy can be found by solving the
156 unconstrained saddle-point optimization problem $\max_{\boldsymbol{\lambda} \in \mathbb{R}^d, \pi \in \Pi} \min_{\boldsymbol{\theta} \in \mathbb{R}^d} f(\boldsymbol{\lambda}, \pi; \boldsymbol{\theta})$. Furthermore,
157 since the optimal policy can be written as $\pi^*(a|x) = \mathbb{I}_{\left\{a = \operatorname{argmax}_b \langle \boldsymbol{\theta}^{\pi^*}, \boldsymbol{\varphi}(x, b) \rangle\right\}}$, it is sufficient to
158 consider softmax policies of the form

$$\Pi(D_\pi) = \left\{ \pi_\theta(a|x) = \frac{e^{\langle \boldsymbol{\varphi}(x, a), \boldsymbol{\theta} \rangle}}{\sum_{a'} e^{\langle \boldsymbol{\varphi}(x, a'), \boldsymbol{\theta} \rangle}} \,\middle|\, \boldsymbol{\theta} \in \mathbb{B}_d(D_\pi) \right\},$$

159 which can approximate $\pi^*$ to good precision when the diameter $D_\pi$ is set to be large enough. This
160 parametrization effectively reduces the high-dimensional LP into a low-dimensional saddle-point
161 optimization problem.

## 3 Feature-occupancy gradient ascent for offline RL in linear MDPs

A natural idea for developing RL methods is to build an empirical approximation of the function $f$ defined in the previous section, and use primal-dual methods to find a saddle-point of the resulting approximation. For offline RL, this approach has been explored by Gabbianelli et al. [2024] and Hong and Tewari [2024]. In this work, we develop an alternative approach that seeks to directly optimize the return by approximately maximizing the unconstrained primal function $f^* : \mathbb{R}^d \times \Pi$, defined for each feature-occupancy vector $\boldsymbol{\lambda}$ and policy $\pi$ as

$$f^*(\boldsymbol{\lambda}, \pi) = \min_{\boldsymbol{\theta} \in \mathbb{B}_d(D_{\boldsymbol{\theta}})} f(\boldsymbol{\lambda}, \pi; \boldsymbol{\theta}),$$

for an appropriately chosen feasible set $\mathbb{B}_d(D_{\boldsymbol{\theta}})$. Given the discussion in the previous section, maximizing this function with respect to $\boldsymbol{\lambda}$ and $\pi$ is rightly expected to result in an optimal policy (which intuition will be made formal in our analysis). Notably, the so-called objective $f$ in Equation (7) depends on the transition weight matrix $\boldsymbol{\Psi}$ which is unknown in general. As we soon show, this matrix dominates the loss of the $\boldsymbol{\theta}$-player and $\boldsymbol{\lambda}$-player. Based on these observations, our approach consists of building a well-chosen estimator $\widehat{f}$ of $f$, and then maximizing the associated primal function $\widehat{f}^*$ defined as

$$\widehat{f}^*(\boldsymbol{\lambda}, \pi) = \min_{\boldsymbol{\theta} \in \mathbb{B}_d(D_{\boldsymbol{\theta}})} \widehat{f}(\boldsymbol{\lambda}; \boldsymbol{\theta}, \pi).$$

The objective $\widehat{f}$ is built via a least-squares estimator inspired by the classic LSTD model estimate of Bradtke and Barto [1996], Parr et al. [2008], which has been successfully used for analyzing finite-horizon linear MDPs in a variety of recent works (e.g., Jin et al., 2020, Neu and Pike-Burke, 2020). In particular, we fit an estimator $\widehat{\boldsymbol{\Psi}}$ of the true matrix $\boldsymbol{\Psi}$ using samples from the dataset $\mathcal{D}_n = \{(X_i, A_i, R_i, X_i')\}_{i=1}^n$ as follows. Let $\boldsymbol{\varphi}_i = \boldsymbol{\varphi}(X_i, A_i)$ denote the feature vector of $(X_i, A_i)$ and $\boldsymbol{\Lambda}_n = \beta \boldsymbol{I}_n + \frac{1}{n} \sum_{i=1}^n \boldsymbol{\varphi}_i \boldsymbol{\varphi}_i^\mathsf{T}$ the empirical feature covariance matrix. We define the regularized least squares estimate of $\boldsymbol{\Psi}$ at $x \in \mathcal{X}$ as

$$\widehat{\boldsymbol{\psi}}(x) = \arg\min_{\boldsymbol{\psi}(x) \in \mathbb{R}^d} \frac{1}{n} \sum_{i=1}^n \left( \langle \boldsymbol{\varphi}_i, \boldsymbol{\psi}(x) \rangle - \mathbb{I}_{\{x = X_i'\}} \right)^2 + \beta \|\boldsymbol{\psi}(x)\|_2^2,$$

so that the estimate can be written as

$$\widehat{\boldsymbol{\Psi}} = \sum_{x \in \mathcal{X}} \widehat{\boldsymbol{\psi}}(x) \boldsymbol{e}_x^\mathsf{T} = \frac{1}{n} \boldsymbol{\Lambda}_n^{-1} \sum_{i=1}^n \boldsymbol{\varphi}_i \boldsymbol{e}_{X_i'}^\mathsf{T}. \tag{8}$$

With this matrix at hand, we define $\widehat{f}$ as

$$\widehat{f}(\boldsymbol{\lambda}, \pi; \boldsymbol{\theta}) = (1 - \gamma)\langle \boldsymbol{\nu}_0, \boldsymbol{v}_{\boldsymbol{\theta}, \pi} \rangle + \langle \boldsymbol{\lambda}, \boldsymbol{\omega} + \gamma \widehat{\boldsymbol{\Psi}} \boldsymbol{v}_{\boldsymbol{\theta}, \pi} - \boldsymbol{\theta} \rangle = \langle \boldsymbol{\lambda}, \boldsymbol{\omega} \rangle + \langle \boldsymbol{\theta}, \boldsymbol{\Phi}^\mathsf{T} \widehat{\boldsymbol{\mu}}_{\boldsymbol{\lambda}, \pi} - \boldsymbol{\lambda} \rangle,$$

where $\widehat{\mu}_{\boldsymbol{\lambda}, \pi}(x, a) = \pi(a|x) \left[ (1 - \gamma)\nu_0(x) + \gamma \langle \widehat{\boldsymbol{\psi}}(x), \boldsymbol{\lambda} \rangle \right]$ is a sample-based approximation of $\mu_{\boldsymbol{\lambda}, \pi}$.

For the purpose of optimization, we will employ appropriately chosen versions of mirror ascent [Nemirovski and Yudin, 1983, Beck and Teboulle, 2003] to iteratively optimize the primal variables. Denoting the iterates for each $t = 1, 2, \ldots, T$ by $\boldsymbol{\lambda}_t$ and $\pi_t$, and defining $\theta_t = \arg\min_{\boldsymbol{\theta} \in \mathbb{B}_d(D_{\boldsymbol{\theta}})} \widehat{f}(\boldsymbol{\lambda}_t, \pi_t; \boldsymbol{\theta})$, the updates are defined as follows. Using $\boldsymbol{g}_{\boldsymbol{\lambda}}(t) = \nabla_{\boldsymbol{\lambda}_t} \widehat{f}^*(\boldsymbol{\lambda}_t, \pi_t)$ to denote the gradient of $\widehat{f}^*$ with respect to the feature occupancies, the first set of variables is updated as

$$\boldsymbol{\lambda}_{t+1} = \arg\max_{\boldsymbol{\lambda} \in \mathbb{R}^d} \left\{ \langle \boldsymbol{\lambda}, \boldsymbol{g}_{\boldsymbol{\lambda}}(t) \rangle - \frac{1}{2\eta} \|\boldsymbol{\lambda} - \boldsymbol{\lambda}_t\|_{\boldsymbol{\Lambda}_n^{-1}}^2 - \frac{\varrho}{2} \|\boldsymbol{\lambda}\|_{\boldsymbol{\Lambda}_n^{-1}}^2 \right\}, \tag{9}$$

where the first regularization term acts as proximal regularization (necessary for mirror-ascent-style methods), and the second one has a stabilization effect whose role will be made clear later in the analysis. The resulting update can be written in closed form, and is equivalent to a preconditioned gradient-ascent step on $\widehat{f}^*$. The policies are updated in each state-action pair $x, a$ as

$$\pi_{t+1}(a|x) = \frac{\pi_t(a|x) e^{\alpha \langle \varphi(x,a), \boldsymbol{\theta}_t \rangle}}{\sum_{a'} \pi_t(a'|x) e^{\alpha \langle \varphi(x,a'), \boldsymbol{\theta}_t \rangle}} = \frac{\pi_1(a|x) e^{\alpha \langle \varphi(x,a), \sum_{k=1}^t \boldsymbol{\theta}_k \rangle}}{\sum_{a'} \pi_1(a'|x) e^{\alpha \langle \varphi(x,a'), \sum_{k=1}^t \boldsymbol{\theta}_k \rangle}},$$

---

**Algorithm 1** Feature-Occupancy Gradient Ascent (`FOGAS`)

---

**Input:** Learning rates $\alpha, \varrho, \eta$, initial points $\boldsymbol{\lambda}_1 \in \mathbb{R}^d, \pi_1 \in \Pi(D_\pi), \bar{\boldsymbol{\theta}}_0 = \mathbf{0}$, and dataset $\mathcal{D}_n$.

**for** $t = 1$ **to** $T$ **do**

    *// Value-parameter update*

    Compute

    $\boldsymbol{\Phi}^\intercal \widehat{\boldsymbol{\mu}}_{\boldsymbol{\lambda}_t, \pi_t} = (1 - \gamma) \sum_a \pi_t(a|x_0) \boldsymbol{\varphi}(x_0, a) + \gamma \frac{1}{n} \sum_{i=1}^n \sum_a \pi_t(a|X'_i) \boldsymbol{\varphi}(X'_i, a) \left\langle \boldsymbol{\varphi}_i, \boldsymbol{\Lambda}_n^{-1} \boldsymbol{\lambda}_t \right\rangle$

    $\boldsymbol{\theta}_t = \arg\min_{\boldsymbol{\theta} \in \mathbb{B}_d(D_\theta)} \langle \boldsymbol{\theta}, \boldsymbol{\Phi}^\intercal \widehat{\boldsymbol{\mu}}_{\boldsymbol{\lambda}_t, \pi_t} - \boldsymbol{\lambda}_t \rangle$

    *// Policy update*

    Update $\bar{\boldsymbol{\theta}}_t = \bar{\boldsymbol{\theta}}_{t-1} + \boldsymbol{\theta}_t$

    $\pi_{t+1} = \sigma\left(\alpha \boldsymbol{\Phi} \bar{\boldsymbol{\theta}}_t\right)$

    *// Feature-occupancy update*

    Compute $\widehat{\boldsymbol{\Psi}} \boldsymbol{v}_{\boldsymbol{\theta}_t, \pi_t} = \frac{1}{n} \boldsymbol{\Lambda}_n^{-1} \sum_{i=1}^n \boldsymbol{\varphi}_i \boldsymbol{v}_{\boldsymbol{\theta}_t, \pi_t}(X'_i)$

    Compute $\boldsymbol{g}_{\boldsymbol{\lambda}}(t) = \boldsymbol{\omega} + \gamma \widehat{\boldsymbol{\Psi}} \boldsymbol{v}_{\boldsymbol{\theta}_t, \pi_t} - \boldsymbol{\theta}_t$

    $\boldsymbol{\lambda}_{t+1} = \frac{1}{1 + \varrho\eta}\left(\boldsymbol{\lambda}_t + \eta \boldsymbol{\Lambda}_n \boldsymbol{g}_{\boldsymbol{\lambda}}(t)\right)$

**end for**

**return** $\pi_J$ with $J \sim \mathcal{U}(1, \cdots, T)$.

---

corresponding to performing an entropy-regularized mirror ascent step in each state $x$ (cf. Neu et al., 2017). We use the shorthand notation $\pi_{t+1} = \sigma\left(\alpha \boldsymbol{\Phi} \sum_{k=1}^t \boldsymbol{\theta}_k\right)$ to denote the resulting softmax policy, and note that it is fully specified by a $d$-dimensional vector that can be stored compactly. After the final iterate is computed, the algorithm picks the index $J$ uniformly at random and outputs the policy $\pi_J$. We refer to the resulting algorithm as *Feature-Occupancy Gradient AScent* (`FOGAS`), and present its detailed pseudocode featuring the explicit expressions of $\boldsymbol{\lambda}_t$ and $\boldsymbol{\theta}_t$ as Algorithm 1.

The following theorem states our main result regarding the performance of `FOGAS`.

**Theorem 3.1.** *Let $\pi_1$ be the uniform policy and $\boldsymbol{\lambda}_1 = \mathbf{0}$. Also set $D_\theta = \sqrt{d}/(1 - \gamma)$, $D_\pi = \alpha T D_\theta$ and $\delta > 0$. Suppose that we run* `FOGAS` *for $T \geq \frac{2R^2 n \log A}{\log(1/\delta)}$ rounds with parameters $\beta = R^2/dT$ as well as*

$$\alpha = \sqrt{\frac{2(1-\gamma)^2 \log A}{R^2 dT}}, \quad \varrho = \gamma \sqrt{\frac{320 d^2 \log(2T/\delta)}{(1-\gamma)^2 n}}, \quad \eta = \sqrt{\frac{(1-\gamma)^2}{27 R^2 d^2 T}}.$$

*Then, with probability at least $1 - \delta$, the following bound is satisfied for any comparator policy $\pi^*$ and the associated feature-occupancy vector $\boldsymbol{\lambda}^{\pi^*} = \boldsymbol{\Phi}^\intercal \boldsymbol{\mu}^{\pi^*}$:*

$$\mathbb{E}_J\left[\left\langle \boldsymbol{\mu}^{\pi^*} - \boldsymbol{\mu}^{\pi_J}, \boldsymbol{r} \right\rangle\right] = \mathcal{O}\left(\frac{\|\boldsymbol{\lambda}^{\pi^*}\|_{\boldsymbol{\Lambda}_n^{-1}}^2 + 1}{1 - \gamma} \cdot \sqrt{\frac{d^2 \log(2T/\delta)}{n}}\right),$$

*with the expectation taken with respect to the random index $J$.*

The most important factor in the bound of Theorem 3.1 is $\|\boldsymbol{\lambda}^*\|_{\boldsymbol{\Lambda}_n^{-1}}^2$, which measures the extent to which the data $\mathcal{D}_n$ covers the comparator policy $\pi^*$ in feature space. We accordingly refer to this quantity as the *feature coverage ratio* between the policy $\pi^*$ and the data set $\mathcal{D}_n$, and we discuss its relationship with other notions of data coverage in Section 5. Notably, the bound holds simultaneously for all comparator policies $\pi^*$, and thus it can be restated in an oracle-inequality form. On the same note, `FOGAS` does not need any prior upper bounds on the comparator norm $\|\boldsymbol{\lambda}^*\|_{\boldsymbol{\Lambda}_n^{-1}}^2$, and in particular it does not project the iterates $\boldsymbol{\lambda}_t$ to a bounded set. These nontrivial properties are enabled by a recently proposed stabilization trick due to Jacobsen and Cutkosky [2023] and Neu and Okolo [2024], which amounts to augmenting the standard mirror-ascent update of Equation (9) with the regularization term $\frac{\varrho}{2}\|\boldsymbol{\lambda}\|_{\boldsymbol{\Lambda}_n^{-1}}^2$. Without this additional regularization, the bounds would feature an additional factor of the order $\frac{1}{T}\sum_{t=1}^T \|\boldsymbol{\lambda}_t\|_{\boldsymbol{\Lambda}_n^{-1}}^2$, which cannot be controlled without projecting the iterates and in any case make it impossible to prove a comparator-adaptive bound. We defer further discussion of the result to Section 5.

## 4 Analysis

This section is dedicated to proving our main result, Theorem 3.1. While we have defined `FOGAS` as a "primal-only" algorithm above, its analysis will be most convenient if we regard it as a primal-dual algorithm with implicitly defined dual updates. In particular, we will view the updates of `FOGAS` as a sequence of steps in a zero sum game between two teams of players: the *max players* that control $\boldsymbol{\lambda}_t$ and $\pi_t$, and the *min player* that picks $\boldsymbol{\theta}_t$. The min player uses the simple *best-response* strategy of picking $\boldsymbol{\theta}_t = \arg\min_{\boldsymbol{\theta} \in \mathbb{B}_d(D_{\boldsymbol{\theta}})} \widehat{f}(\boldsymbol{\lambda}, \pi_t)$, and the other two players perform their updates via appropriate versions mirror ascent on their respective objectives. Importantly, the updates of the $\boldsymbol{\lambda}$-player are based on the gradients of $\widehat{f}^*$, which satisfy

$$\boldsymbol{g}_{\boldsymbol{\lambda}}(t) = \nabla_{\boldsymbol{\lambda}_t} \widehat{f}^*(\boldsymbol{\lambda}_t, \pi_t) = \nabla_{\boldsymbol{\lambda}_t} \left( \min_{\boldsymbol{\theta} \in \mathbb{B}_d(D_{\boldsymbol{\theta}})} \widehat{f}(\boldsymbol{\lambda}_t, \pi_t; \boldsymbol{\theta}) \right) = \nabla_{\boldsymbol{\lambda}_t} \widehat{f}(\boldsymbol{\lambda}_t, \pi_t; \boldsymbol{\theta}_t),$$

where the last equality follows from an application of Danskin's theorem. This property enables a major conceptual simplification that allows the interpretation of the updates as optimizing the unconstrained primal $\widehat{f}^*$ directly. We refer the interested reader to Chapter 6 of Bertsekas [1997] for more context on such use of primal-dual analysis.

More concretely, we make use of an analysis technique first developed by Neu and Okolo [2023], and further refined by Gabbianelli et al. [2024] and Hong and Tewari [2024]. The core idea is to introduce the *dynamic duality gap* defined on a sequence of iterates $\{(\boldsymbol{\lambda}_t, \pi_t, \boldsymbol{\theta}_t)\}_{t=1}^{T}$ produced by some iterative method, and a set of well-chosen *comparators* $\left(\boldsymbol{\lambda}^*, \pi^*; \{\boldsymbol{\theta}_t^*\}_{t=1}^{T}\right)$ as

$$\mathfrak{G}_T \left(\boldsymbol{\lambda}^*, \pi^*; \{\boldsymbol{\theta}_t^*\}_{t=1}^{T}\right) = \frac{1}{T} \sum_{t=1}^{T} \left( f(\boldsymbol{\lambda}^*, \pi^*; \boldsymbol{\theta}_t) - f(\boldsymbol{\lambda}_t, \pi_t; \boldsymbol{\theta}_t^*) \right).$$

Similar to Lemma 4.1 of Gabbianelli et al. [2024], we show in Lemma 4.1 below that with an appropriate choice of the comparator points, we can relate the gap to the expected suboptimality of policy $\pi_J$ where $J \sim \mathcal{U}(1, \cdots, T)$. We leave the proof in Appendix B.1.1.

**Lemma 4.1.** *Suppose that* $D_{\boldsymbol{\theta}} = \sqrt{d}/(1-\gamma)$. *Choose* $(\boldsymbol{\lambda}^*, \pi^*, \boldsymbol{\theta}_t^*) = \left(\boldsymbol{\Phi}^{\mathsf{T}} \boldsymbol{\mu}^{\pi^*}, \pi^*, \boldsymbol{\theta}^{\pi_t}\right) \in \mathbb{R}^d \times \Pi(D_{\pi}) \times \mathbb{B}_d(D_{\theta})$ *for* $t = 1, \cdots, T$ *where* $\boldsymbol{\mu}^{\pi^*}$ *is a valid occupancy measure induced by* $\pi^*$. *Then,*

$$\mathbb{E}_J \left[ \left\langle \boldsymbol{\mu}^{\pi^*} - \boldsymbol{\mu}^{\pi_J}, \boldsymbol{r} \right\rangle \right] = \mathfrak{G}_T \left( \boldsymbol{\Phi}^{\mathsf{T}} \boldsymbol{\mu}^{\pi^*}, \pi^*, \{\boldsymbol{\theta}^{\pi_t}\}_{t=1}^{T} \right).$$

We will show below that the dynamic duality gap can be written in terms of the *regrets* of each player and an additional term related to the estimation error of $\widehat{f}$, and then proceed to provide bounds on all of these quantities. Specifically, the regrets of each player with respect to each of their respective comparators are defined as

$$\mathfrak{R}_T(\pi^*) = \sum_{t=1}^{T} \sum_x \nu^*(x) \sum_a (\pi^*(a|x) - \pi_t(a|x)) q_t(x, a),$$

$$\mathfrak{R}_T(\boldsymbol{\lambda}^*) = \sum_{t=1}^{T} \widehat{f}(\boldsymbol{\lambda}^*, \pi_t; \boldsymbol{\theta}_t) - \widehat{f}(\boldsymbol{\lambda}_t, \pi_t; \boldsymbol{\theta}_t) = \sum_{t=1}^{T} \langle \boldsymbol{\lambda}^* - \boldsymbol{\lambda}_t, \boldsymbol{\omega} + \gamma \widehat{\boldsymbol{\Psi}} \boldsymbol{v}_{\boldsymbol{\theta}_t, \pi_t} - \boldsymbol{\theta}_t \rangle,$$

$$\mathfrak{R}_T(\boldsymbol{\theta}_{1:T}^*) = \sum_{t=1}^{T} \widehat{f}(\boldsymbol{\lambda}_t, \pi_t; \boldsymbol{\theta}_t) - \widehat{f}(\boldsymbol{\lambda}_t, \pi_t; \boldsymbol{\theta}_t^*) = \sum_{t=1}^{T} \langle \boldsymbol{\theta}_t - \boldsymbol{\theta}_t^*, \boldsymbol{\Phi}^{\mathsf{T}} \widehat{\boldsymbol{\mu}}_{\boldsymbol{\lambda}_t, \pi_t} - \boldsymbol{\lambda}_t \rangle.$$

where $\boldsymbol{\nu}^* = (1-\gamma)\nu_0(x) + \gamma \langle \psi(x), \boldsymbol{\lambda}^* \rangle$. Furthermore, we define the *gap-estimation error* as

$$\text{err}_{\widehat{\boldsymbol{\Psi}}} = \sum_{t=1}^{T} \langle \boldsymbol{\lambda}^*, \left(\boldsymbol{\Psi} - \widehat{\boldsymbol{\Psi}}\right) \boldsymbol{v}_{\boldsymbol{\theta}_t, \pi_t} \rangle + \sum_{t=1}^{T} \langle \boldsymbol{\lambda}_t, \left(\widehat{\boldsymbol{\Psi}} - \boldsymbol{\Psi}\right) \boldsymbol{v}_{\boldsymbol{\theta}_t^*, \pi_t} \rangle. \tag{10}$$

The following lemma rewrites the duality gap using the above terms.

**Lemma 4.2.** *The dynamic duality gap satisfies*

$$\mathfrak{G}_T(\boldsymbol{\lambda}^*, \pi^*, \boldsymbol{\theta}_{1:T}^*) = \frac{1}{T} \mathfrak{R}_T(\pi^*) + \frac{1}{T} \mathfrak{R}_T(\boldsymbol{\lambda}^*) + \frac{1}{T} \mathfrak{R}_T(\boldsymbol{\theta}_{1:T}^*) + \frac{\gamma}{T} err_{\widehat{\boldsymbol{\Psi}}}.$$

The proof directly follows from a straightforward calculation similar to the proof of Lemma 4.2 of Gabbianelli et al. [2024] and Section E.1 of Hong and Tewari [2024] which is reproduced in Appendix B.1.2 for completeness. It remains to bound the regret of the players, as well as the gap-estimation error. An obstacle we need to face in the analysis is that our bound of the latter error term scale with $\frac{1}{T}\sum_{t=1}^{T}\|\boldsymbol{\lambda}_t\|_{\boldsymbol{\Lambda}_n^{-1}}^2$, which is undesirable given our aspiration to achieve bounds that scale only with the comparator norm $\|\boldsymbol{\lambda}^*\|_{\boldsymbol{\Lambda}_n^{-1}}^2$ without requiring prior upper bounds on this quantity (that would enable us to project the iterates to a bounded domain). This challenge is addressed by making use of the stabilization technique of Jacobsen and Cutkosky [2023] and Neu and Okolo [2024] in the updates for the $\boldsymbol{\lambda}$-player, which effectively eliminates these problematic terms. We briefly outline the remaining parts of the analysis below.

## 4.1 Regret analysis

The regrets of each player are respectively controlled by the following three lemmas.

**Lemma 4.3.** *Suppose that $\boldsymbol{\nu}^* \in \Delta_\mathcal{X}$. Let $\pi_1$ be the uniform policy which selects all actions with equal probability in each state. Under the conditions on the feature map in Definition 2.1, the regret of the $\pi$-player against $\pi^*$ satisfies $\frac{1}{T}\mathfrak{R}_T\left(\pi^*\right) \leq \frac{\log A}{\alpha T} + \frac{\alpha R^2 D_{\boldsymbol{\theta}}^2}{2}$.*

The proof is a standard application of the analysis of exponential-weight updates, stated as Lemma E.1.

**Lemma 4.4.** *Let $\boldsymbol{\lambda}_1 = \mathbf{0}$ and $C = 6\beta\left(d + D_{\boldsymbol{\theta}}^2\right) + 3d\left(1 + RD_{\boldsymbol{\theta}}\right)^2 + 3\gamma^2 dR^2 D_{\boldsymbol{\theta}}^2$. Then, the regret of the $\boldsymbol{\lambda}$-player against any comparator $\boldsymbol{\lambda}^* \in \mathbb{R}^d$ satisfies*

$$\frac{1}{T}\mathfrak{R}_T\left(\boldsymbol{\lambda}^*\right) \leq \left(\frac{1}{2\eta T} + \frac{\varrho}{2}\right)\|\boldsymbol{\lambda}^*\|_{\boldsymbol{\Lambda}_n^{-1}}^2 + \frac{\eta C}{2} - \frac{\varrho}{2T}\sum_{t=1}^{T}\|\boldsymbol{\lambda}_t\|_{\boldsymbol{\Lambda}_n^{-1}}^2 .$$

The proof (provided in Appendix B.2.2) follows from applying the standard analysis of composite-objective mirror descent due to Duchi et al. [2010] (stated as Lemma C.1 in the Appendix) and the bound $\|\boldsymbol{\Lambda}_n \boldsymbol{g}_{\boldsymbol{\lambda}}(t)\|_{\boldsymbol{\Lambda}_n^{-1}}^2 \leq C$ on the weighted norm of the gradients for all $t$ provided in Lemma C.2.

**Lemma 4.5.** *Let $D_{\boldsymbol{\theta}} = \sqrt{d}/\left(1 - \gamma\right)$. The regret of the $\theta$-player satisfies $\frac{1}{T}\mathfrak{R}_T\left(\boldsymbol{\theta}_{1:T}^*\right) \leq 0$.*

As we show in Appendix B.2.3, the above statement holds trivially thanks to the "best-response" definition of $\theta_t$. This concludes our regret analysis.

## 4.2 Bounding the gap-estimation error

The following statement (proved in Appendix B.3) provides a bound on $\mathrm{err}_{\widehat{\boldsymbol{\Psi}}}$:

**Lemma 4.6.** *Suppose that $\|\boldsymbol{\varphi}(x,a)\|_2 \leq R$ for all $(x,a) \in \mathcal{X} \times \mathcal{A}$, $D_{\boldsymbol{\theta}} = \sqrt{d}/\left(1-\gamma\right)$ and $\alpha = \sqrt{2\left(1-\gamma\right)^2 \log A / R^2 dT}$ to optimize $\mathfrak{R}_T\left(\pi^*\right)$. Then, for any $T \geq \frac{2R^2 n \log A}{\log(1/\delta)}$ and and $\xi \geq 0$, the following holds with probability at least $1 - \delta$:*

$$\mathrm{err}_{\widehat{\boldsymbol{\Psi}}} \leq \frac{1}{2\xi}\left(\|\boldsymbol{\lambda}^*\|_{\boldsymbol{\Lambda}_n^{-1}}^2 + \frac{1}{T}\sum_{t=1}^{T}\|\boldsymbol{\lambda}_t\|_{\boldsymbol{\Lambda}_n^{-1}}^2\right) + T^2\xi\left(\frac{320d^2 \log\left(2T/\delta\right)}{n\left(1-\gamma\right)^2}\right).$$

## 4.3 The proof of Theorem 3.1

The proof follows from applying Lemmas 4.1 and Lemma 4.2 when $(\boldsymbol{\lambda}^*, \pi^*, \boldsymbol{\theta}_t^*) = \left(\boldsymbol{\Phi}^\top \boldsymbol{\mu}^{\pi^*}, \pi^*, \boldsymbol{\theta}^{\pi_t}\right) \in \mathbb{R}^d \times \Pi\left(D_\pi\right) \times \mathbb{B}_d(D_{\boldsymbol{\theta}})$ for $t = 1, \cdots, T$. Then, adding up the bounds stated in Lemmas 4.3–4.6 under the respective conditions, yields

$$\mathbb{E}_J\left[\left\langle \boldsymbol{\mu}^{\pi^*} - \boldsymbol{\mu}^{\pi_J}, \boldsymbol{r}\right\rangle\right] \leq \sqrt{\frac{d\log\left(1/\delta\right)}{n\left(1-\gamma\right)^2}} + \left(\frac{1}{2\eta T} + \frac{\varrho}{2} + \frac{\gamma}{2\xi T}\right)\left\|\boldsymbol{\lambda}^{\pi^*}\right\|_{\boldsymbol{\Lambda}_n^{-1}}^2 + \frac{\eta C}{2}$$

$$+ \left(\frac{\gamma}{\xi T} - \varrho\right)\frac{1}{2T}\sum_{t=1}^{T}\|\boldsymbol{\lambda}_t\|_{\boldsymbol{\Lambda}_n^{-1}}^2 + \gamma T\xi\left(\frac{320d^2 \log\left(2T/\delta\right)}{n\left(1-\gamma\right)^2}\right).$$

Then, setting $\rho = \frac{\gamma}{\xi T}$ simplifies the second term and eliminates the third term. The claim then follows after optimizing the hyperparameters, with the full details provided in Appendix B.4.

# 5 Discussion

We discuss various aspects of our results below.

**Relation with previous work.** As discussed in the introduction, our work draws heavily on previous contributions of Gabbianelli et al. [2024] and Hong and Tewari [2024]. In particular, our idea of building a least-squares estimator of the transition function is directly borrowed from the latter of these works, and our implicit update rule for $\theta_t$ is also inspired by their work to a good extent. Their approach, however, failed to reach the same degree of efficiency due to a number of suboptimal design choices. First, they used an alternative parametrization of the feature occupancies which only allowed them to work under a more restrictive coverage condition, so that their bounds depend on $\|\boldsymbol{\lambda}^*\|_{\boldsymbol{\Lambda}_n^{-2}}$ which can be much larger than the feature coverage ratio appearing in our bounds. Second, their algorithm required a prior upper bound on this coverage parameter, with the guarantees scaling with the bound rather than the actual coverage. Such bounds are typically difficult to obtain in practice. Third, the implementation of their algorithm required intricate computational steps necessitated by their feature-occupancy parametrization. Our work has successfully removed these limitations and reduced the complexity of their method, thanks to a new primal-only analysis style that we hope will find further uses in reinforcement learning.

**Computational and statistical efficiency.** As can be inferred from our main result, the sample complexity of finding an $\varepsilon$-optimal policy using our algorithm is of the order $d^2 \|\boldsymbol{\lambda}^*\|_{\boldsymbol{\Lambda}_n^{-1}}^2 / \varepsilon^2 (1 - \gamma)^2$, which is optimal in terms of scaling with $\varepsilon$. The rate can be improved to scale linearly with the feature coverage ratio $\|\boldsymbol{\lambda}^*\|_{\boldsymbol{\Lambda}_n^{-1}}$, if a tight upper bound is known on it which can be used for hyperparameter tuning. We find this scenario to be unlikely, and are curious to see if future work can attain this improved scaling without such prior knowledge. As for computational complexity, we point out that the cost of each iteration of our method scales linearly with the sample size $n$, due to having to compute the matrix-vector products $\widehat{\boldsymbol{\Psi}} \boldsymbol{v}_{\boldsymbol{\theta}_t, \pi_t}$. Indeed, the matrix $\widehat{\boldsymbol{\Psi}}$ is sparse with $n$ non-zero rows, and as such computing this product takes linear time in $n$. Since the iteration complexity of FOGAS scales linearly with the sample size $n$, this makes for an overall runtime complexity of order $n^2$. This limitation is of course shared with all methods using the same least-squares transition estimator for the transition model, including all work that builds on Jin et al. [2020], but we nevertheless wonder if a substantial improvement is possible on this front.

**Data coverage assumptions.** The only works we are aware of that scale with the feature-coverage ratio $\|\boldsymbol{\lambda}^*\|_{\boldsymbol{\Lambda}_n^{-1}}$ are due to Zanette et al. [2021] and Gabbianelli et al. [2024]. The latter work only achieves this bound under the assumption that the data is drawn i.i.d. from a fixed behavior policy with known feature covariance matrix, which is a much more restricted setting that we consider here. Such assumptions are not needed by Zanette et al. [2021], however their results are restricted to the simpler finite-horizon MDP setting, and their algorithm is arguably more complex than ours. Using our notation, their approach can be interpreted as solving a "pessimistic" version of the the relaxed dual LP (4) that features some additional quadratic constraints. This approach is not computationally viable for the infinite-horizon discounted case we consider, as it requires solving a fixed-point equation with respect to the estimated transition operator (cf. Wei et al., 2021).

**Possible extensions.** Our approach can be extended and generalized in a variety of ways. First, following Gabbianelli et al. [2024], we believe that it is straightforward to extend our analysis to undiscounted infinite-horizon MDPs. Second, we similarly believe that an extension to constrained MDPs is possible without major challenges, following Hong and Tewari [2024]. We did not pursue these extensions because we believe that they add little additional insight. There are other potential directions that we did not explore because we found them to be too ambitious for the moment. These include extending our results beyond linear MDPs to other MDP models with linear function approximation, including MDPs with low inherent Bellman rank (which may be within reach of the current theory, c.f. Zanette et al., 2020), linearly $Q^\pi$-realizable MDPs (which are known to be challenging, c.f. Weisz et al., 2022, 2024). Even more ambitiously, one can ask if it is possible to extend our methods to work under more general notions of function approximation. This looks very challenging given the central role of feature occupancies in our formalism, which are strictly tied to linear function approximation. We are nevertheless optimistic that the ideas presented in this work will find use in other contexts, possibly including nonlinear function approximation in the future.

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

# Appendix

## A   Missing proofs of Section 2

### A.1   Properties of the relaxed LP

In this section we prove a basic result about the feasible sets of the relaxed linear programs defined in Equations (3) and (4). We remark that similar results have been previously shown in Proposition 4 of Bas-Serrano et al. [2021] and Appendix A.1 of Neu and Okolo [2023].

**Lemma A.1.** *Suppose that the MDP satisfies the linear MDP assumption in the sense of Definition 2.1, consider the relaxed linear programs 3 and 4 and their respective feasible sets:*

$$\mathcal{M}_{\mathbf{\Phi}}^P = \left\{ (\boldsymbol{\lambda}, \boldsymbol{\mu}) \in \mathbb{R}^d \times \mathbb{R}_+^{XA} \mid \boldsymbol{E}^\mathsf{T}\boldsymbol{\mu} = (1-\gamma)\boldsymbol{\nu}_0 + \gamma\boldsymbol{\Psi}^\mathsf{T}\boldsymbol{\lambda}, \quad \boldsymbol{\lambda} = \boldsymbol{\Phi}^\mathsf{T}\boldsymbol{\mu} \right\},$$

$$\mathcal{M}_{\mathbf{\Phi}}^D = \left\{ (\boldsymbol{v}, \boldsymbol{\theta}) \in \mathbb{R}^X \times \mathbb{R}^d \mid \boldsymbol{E}\boldsymbol{v} \geq \boldsymbol{\Phi}\boldsymbol{\theta}, \quad \boldsymbol{\theta} = \boldsymbol{\omega} + \gamma\boldsymbol{\Psi}\boldsymbol{v} \right\}.$$

*Then, the following statements hold:*

- *The set $\mathcal{M} = \left\{ \boldsymbol{\mu} : (\boldsymbol{\lambda}, \boldsymbol{\mu}) \in \mathcal{M}_{\mathbf{\Phi}}^P \right\}$ coincides with the feasible set of the primal LP (1). Furthermore, for all $(\boldsymbol{\lambda}^*, \boldsymbol{\mu}^*) \in \arg\max_{(\boldsymbol{\lambda}, \boldsymbol{\mu}) \in \mathcal{M}_{\mathbf{\Phi}}^P} \langle \boldsymbol{\lambda}, \boldsymbol{\omega} \rangle$, we have that $\boldsymbol{\mu}^*$ is the occupancy measure of an optimal policy.*

- *The set $\mathcal{V} = \left\{ \boldsymbol{v} : (\boldsymbol{v}, \boldsymbol{\theta}) \in \mathcal{M}_{\mathbf{\Phi}}^D \right\}$ coincides with the feasible set of the dual LP (2). Furthermore, the optimal value function $\boldsymbol{v}^{\pi^*}$ and the parameter vector $\boldsymbol{\theta}^{\pi^*}$ satisfying $\boldsymbol{q}^{\pi^*} = \boldsymbol{\Phi}\boldsymbol{\theta}^{\pi^*}$ satisfy $(\boldsymbol{v}^{\pi^*}, \boldsymbol{\theta}^{\pi^*}) \in \arg\min_{(\boldsymbol{v}, \boldsymbol{\theta}) \in \mathcal{M}_{\mathbf{\Phi}}^D} (1-\gamma)\langle \boldsymbol{\nu}_0, \boldsymbol{v} \rangle.$*

*Proof.* We first show that for any feasible point $\boldsymbol{\mu}$ of the LP (1), the tuple $(\boldsymbol{\lambda}, \boldsymbol{\mu})$ is feasible for the relaxed LP with $\boldsymbol{\lambda} = \boldsymbol{\Phi}^\mathsf{T}\boldsymbol{\mu}$. This choice of $\boldsymbol{\lambda}$ satisfies the second primal constraint by definition, so it remains to verify that the first constraint is also satisfied. Indeed, this follows from

$$\boldsymbol{E}^\mathsf{T}\boldsymbol{\mu} - (1-\gamma)\boldsymbol{\nu}_0 - \gamma\boldsymbol{\Psi}^\mathsf{T}\boldsymbol{\lambda} = \boldsymbol{E}^\mathsf{T}\boldsymbol{\mu} - (1-\gamma)\boldsymbol{\nu}_0 - \gamma\boldsymbol{\Psi}^\mathsf{T}\boldsymbol{\Phi}^\mathsf{T}\boldsymbol{\mu}$$
$$= \boldsymbol{E}^\mathsf{T}\boldsymbol{\mu} - (1-\gamma)\boldsymbol{\nu}_0 - \gamma\boldsymbol{P}^\mathsf{T}\boldsymbol{\mu} = 0,$$

where we have used the linear MDP property to write $\boldsymbol{\Psi}^\mathsf{T}\boldsymbol{\Phi}^\mathsf{T} = \boldsymbol{P}^\mathsf{T}$ in the first step and that $\boldsymbol{\mu}$ is a valid occupancy measure in the last one. Conversely, supposing that $(\boldsymbol{\lambda}, \boldsymbol{\mu}) \in \mathcal{M}_{\mathbf{\Phi}}^P$ are feasible for the relaxed LP, we have that

$$\boldsymbol{E}^\mathsf{T}\boldsymbol{\mu} - (1-\gamma)\boldsymbol{\nu}_0 - \gamma\boldsymbol{P}^\mathsf{T}\boldsymbol{\mu} = \boldsymbol{E}^\mathsf{T}\boldsymbol{\mu} - (1-\gamma)\boldsymbol{\nu}_0 - \gamma\boldsymbol{\Psi}^\mathsf{T}\boldsymbol{\Phi}^\mathsf{T}\boldsymbol{\mu}$$
$$= \boldsymbol{E}^\mathsf{T}\boldsymbol{\mu} - (1-\gamma)\boldsymbol{\nu}_0 - \gamma\boldsymbol{\Psi}^\mathsf{T}\boldsymbol{\lambda} = 0,$$

thus verifying that $\boldsymbol{\mu}$ is indeed a valid occupancy measure. Optimality of $(\boldsymbol{\lambda}^*, \boldsymbol{\mu}^*)$ follows from the fact that for any $(\boldsymbol{\lambda}, \boldsymbol{\mu}) \in \mathcal{M}_{\mathbf{\Phi}}^P$, we can write the LP objective as $\langle \boldsymbol{\lambda}, \boldsymbol{\omega} \rangle = \langle \boldsymbol{\mu}, \boldsymbol{r} \rangle$ by the linear MDP assumption, and the standard fact that any solution $\boldsymbol{\mu}^*$ to the primal LP 1 is the occupancy measure of an optimal policy (cf. Theorem 6.9.4 in Puterman, 1994). This concludes the first part of the proof.

For the second part of the proof, let us first consider a feasible solution $\boldsymbol{v}$ for the original dual LP (2). Then, the choice $\boldsymbol{\theta} = \omega + \gamma\Psi\boldsymbol{v}$ satisfies the second dual constraint by definition. The first constraint can be verified by writing

$$E\boldsymbol{v} - \Phi\theta = E\boldsymbol{v} - r - \gamma P\boldsymbol{v} \geq 0,$$

where we used the choice of $\boldsymbol{\theta}$ in the first step and the feasibility of $\boldsymbol{v}$ for the original LP in the second step. Conversely, supposing that $(\boldsymbol{\theta}, \boldsymbol{v}) \in \mathcal{M}_{\mathbf{\Phi}}^D$, we note that

$$E\boldsymbol{v} - r - \gamma P\boldsymbol{v} = E\boldsymbol{v} - \Phi\boldsymbol{\theta} \geq 0,$$

which implies the feasibility of $\boldsymbol{v}$ in the LP 2. Optimality of $\boldsymbol{v}^*$ for both LPs follows from the fact that their objectives are identical, and the standard fact that $\boldsymbol{v}^*$ is an optimal solution of the dual LP (2) (cf. Theorem 6.2.2 in Puterman, 1994). $\qquad\square$

# B Missing proofs of Section 4

In this section, we provide performance guarantees for Algorithm 1 in terms of the expected suboptimality of the output policy $\pi_J$, and in particular prove the lemmas provided in Section 4 in the main text. Auxiliary lemmas and technical results for proving some of these are included in Appendix E.

## B.1 Properties of the Dynamic Duality Gap

We first prove our claims regarding the dynamic duality gap introduced in Section 4 of the main text. First, we relate the gap to the expected suboptimality (in terms of return) of $\pi_J$ against a comparator policy $\pi^*$ in Appendix B.1.1. Next, we relate the dynamic duality gap to the average regret of each player in Appendix B.1.2.

### B.1.1 Proof of Lemma 4.1

By definition of the the dynamic duality gap, we have that

$$\mathfrak{G}_T\left(\mathbf{\Phi}^{\mathsf{T}}\boldsymbol{\mu}^{\pi^*}, \pi^*, \{\boldsymbol{\theta}^{\pi_t}\}_{t=1}^T\right) = \frac{1}{T}\sum_{t=1}^T f(\mathbf{\Phi}^{\mathsf{T}}\boldsymbol{\mu}^{\pi^*}, \pi^*; \boldsymbol{\theta}_t) - f(\boldsymbol{\lambda}_t, \pi_t; \boldsymbol{\theta}^{\pi_t}).$$

Considering the first term, we see that

$$\begin{aligned}
f(\mathbf{\Phi}^{\mathsf{T}}\boldsymbol{\mu}^{\pi^*}, \pi^*; \boldsymbol{\theta}_t) &= \left\langle \mathbf{\Phi}^{\mathsf{T}}\boldsymbol{\mu}^{\pi^*}, \boldsymbol{\omega} \right\rangle + \left\langle \boldsymbol{\theta}_t, \mathbf{\Phi}^{\mathsf{T}}\boldsymbol{\mu}_{\boldsymbol{\lambda}^*,\pi^*} - \mathbf{\Phi}^{\mathsf{T}}\boldsymbol{\mu}^{\pi^*} \right\rangle \\
&\overset{(a)}{=} \left\langle \boldsymbol{\mu}^{\pi^*}, \boldsymbol{r} \right\rangle + \left\langle \boldsymbol{\theta}_t, \mathbf{\Phi}^{\mathsf{T}}\boldsymbol{\mu}_{\boldsymbol{\lambda}^*,\pi^*} - \mathbf{\Phi}^{\mathsf{T}}\boldsymbol{\mu}^{\pi^*} \right\rangle \\
&\overset{(b)}{=} \left\langle \boldsymbol{\mu}^{\pi^*}, \boldsymbol{r} \right\rangle,
\end{aligned}$$

where we have used $(a)$ the linear MDP property (definition 2.1) and $(b)$ the following relation:

$$\begin{aligned}
\mu_{\boldsymbol{\lambda}^*,\pi^*}(x,a) &= \pi^*(a|x)\Big[(1-\gamma)\nu_0(x) + \gamma\left\langle \boldsymbol{\psi}(x), \mathbf{\Phi}^{\mathsf{T}}\boldsymbol{\mu}^{\pi^*} \right\rangle\Big] \\
&= \pi^*(a|x)\Big[(1-\gamma)\nu_0(x) + \gamma\sum_{x',a'} p\left(x|x',a'\right)\mu^{\pi^*}\left(x',a'\right)\Big] = \mu^{\pi^*}(x,a).
\end{aligned}$$

Now for the second term, we have

$$\begin{aligned}
f(\boldsymbol{\lambda}_t, \pi_t; \boldsymbol{\theta}^{\pi_t}) &= (1-\gamma)\left\langle \boldsymbol{\nu}_0, \boldsymbol{v}_{\boldsymbol{\theta}^{\pi_t},\pi_t} \right\rangle + \left\langle \boldsymbol{\lambda}_t, \boldsymbol{\omega} + \gamma\mathbf{\Psi}\boldsymbol{v}_{\boldsymbol{\theta}^{\pi_t},\pi_t} - \boldsymbol{\theta}^{\pi_t} \right\rangle \\
&= \left\langle \boldsymbol{\mu}^{\pi_t}, \boldsymbol{r} \right\rangle + \left\langle \boldsymbol{\lambda}_t, \boldsymbol{\omega} + \gamma\mathbf{\Psi}\boldsymbol{v}^{\pi_t} - \boldsymbol{\theta}^{\pi_t} \right\rangle \\
&= \left\langle \boldsymbol{\mu}^{\pi_t}, \boldsymbol{r} \right\rangle,
\end{aligned}$$

where we have used the Bellman equations $\boldsymbol{q}^{\pi_t} = \mathbf{\Phi}\boldsymbol{\theta}_t^{\pi} = \boldsymbol{r} + \gamma\boldsymbol{P}\boldsymbol{v}^{\pi_t} = \mathbf{\Phi}\left(\boldsymbol{\omega} + \gamma\mathbf{\Psi}\boldsymbol{v}^{\pi_t}\right)$, which together with the fact that $\mathbf{\Phi}$ is full rank implies that $\boldsymbol{\theta}^{\pi_t} = \boldsymbol{\omega} + \gamma\mathbf{\Psi}\boldsymbol{v}^{\pi_t}$. Substituting the above expressions for $f(\mathbf{\Phi}^{\mathsf{T}}\boldsymbol{\mu}^{\pi^*}, \pi^*; \boldsymbol{\theta}_t)$ and $f(\boldsymbol{\lambda}_t, \pi_t; \boldsymbol{\theta}^{\pi_t})$ in the dynamic duality gap and noting that $\pi_J$ is such that $\frac{1}{T}\sum_{t=1}^T \left\langle \boldsymbol{\mu}^{\pi_t}, \boldsymbol{r} \right\rangle = \mathbb{E}_J\left[\left\langle \boldsymbol{\mu}^{\pi_J}, \boldsymbol{r} \right\rangle\right]$ we get

$$\mathfrak{G}_T\left(\mathbf{\Phi}^{\mathsf{T}}\boldsymbol{\mu}^{\pi^*}, \pi^*, \{\boldsymbol{\theta}^{\pi_t}\}_{t=1}^T\right) = \mathbb{E}_J\left[\left\langle \boldsymbol{\mu}^{\pi^*} - \boldsymbol{\mu}^{\pi_J}, \boldsymbol{r} \right\rangle\right].$$

This completes the proof. $\qquad\square$

### B.1.2 Proof of Lemma 4.2

Recall that for any comparator points $\left(\boldsymbol{\lambda}^*, \pi^*; \{\boldsymbol{\theta}_t^*\}_{t=1}^T\right)$, the dynamic duality gap is defined as

$$\mathfrak{G}_T\left(\boldsymbol{\lambda}^*, \pi^*; \{\boldsymbol{\theta}_t^*\}_{t=1}^T\right) = \frac{1}{T}\sum_{t=1}^T \left(f(\boldsymbol{\lambda}^*, \pi^*; \boldsymbol{\theta}_t) - f(\boldsymbol{\lambda}_t, \pi_t; \boldsymbol{\theta}_t^*)\right).$$

491 Then, by adding and subtracting some terms we express the dynamic duality gap in terms of the
492 average loss of each player with respect to the objective $f(\boldsymbol{\lambda}, \pi; \boldsymbol{\theta})$. This gives

$$\mathfrak{G}_T(\boldsymbol{\lambda}^*, \pi^*, \boldsymbol{\theta}_{1:T}^*) = \frac{1}{T} \sum_{t=1}^{T} f(\boldsymbol{\lambda}^*, \pi^*; \boldsymbol{\theta}_t) - f(\boldsymbol{\lambda}^*, \pi_t; \boldsymbol{\theta}_t)$$

$$+ \frac{1}{T} \sum_{t=1}^{T} f(\boldsymbol{\lambda}^*, \pi_t; \boldsymbol{\theta}_t) - f(\boldsymbol{\lambda}_t, \pi_t; \boldsymbol{\theta}_t)$$

$$+ \frac{1}{T} \sum_{t=1}^{T} f(\boldsymbol{\lambda}_t, \pi_t; \boldsymbol{\theta}_t) - f(\boldsymbol{\lambda}_t, \pi_t; \boldsymbol{\theta}_t^*). \tag{11}$$

493 Consider the first set of terms from the above expression. By definition of $f$ in Equation (7), we
494 immediately obtain the *instantaneous* regret of the $\pi$-player as

$$f(\boldsymbol{\lambda}^*, \pi^*; \boldsymbol{\theta}_t) - f(\boldsymbol{\lambda}^*, \pi_t; \boldsymbol{\theta}_t) = \langle \boldsymbol{\theta}_t, \boldsymbol{\Phi}^\mathsf{T} \boldsymbol{\mu}_{\boldsymbol{\lambda}^*, \pi^*} - \boldsymbol{\Phi}^\mathsf{T} \boldsymbol{\mu}_{\boldsymbol{\lambda}^*, \pi_t} \rangle$$

$$= \sum_x \nu^*(x) \sum_a (\pi^*(a|x) - \pi_t(a|x)) q_t(x, a)$$

$$= \sum_x \nu^*(x) \sum_a (\pi^*(a|x) - \pi_t(a|x)) q_t(x, a),$$

495 where $\nu^*(x) = (1 - \gamma)\nu_0(x) + \gamma \langle \boldsymbol{\psi}(x), \boldsymbol{\lambda}^* \rangle$. For the regret of the $\boldsymbol{\lambda}$ and $\boldsymbol{\theta}$-players, notice that we
496 can express the estimator $\widehat{f}$ in terms of the objective $f$ as follows:

$$\widehat{f}(\boldsymbol{\lambda}, \pi; \boldsymbol{\theta}) = (1 - \gamma) \langle \boldsymbol{\nu}_0, \boldsymbol{v}_{\boldsymbol{\theta}, \pi} \rangle + \left\langle \boldsymbol{\lambda}, \boldsymbol{\omega} + \gamma \widehat{\boldsymbol{\Psi}} \boldsymbol{v}_{\boldsymbol{\theta}, \pi} - \boldsymbol{\theta} \right\rangle$$

$$= f(\boldsymbol{\lambda}, \pi; \boldsymbol{\theta}) + \gamma \left\langle \boldsymbol{\lambda}, \widehat{\boldsymbol{\Psi}} \boldsymbol{v}_{\boldsymbol{\theta}, \pi} - \boldsymbol{\Psi} \boldsymbol{v}_{\boldsymbol{\theta}, \pi} \right\rangle.$$

497 Taking advantage of this relation, we now consider the last two set of terms in Equation (11). Indeed,
498 for the second set of terms in the equation, we write

$$f(\boldsymbol{\lambda}^*, \pi_t; \boldsymbol{\theta}_t) - f(\boldsymbol{\lambda}_t, \pi_t; \boldsymbol{\theta}_t)$$

$$= \widehat{f}(\boldsymbol{\lambda}^*, \pi_t; \boldsymbol{\theta}_t) - \widehat{f}(\boldsymbol{\lambda}_t, \pi_t; \boldsymbol{\theta}_t) - \gamma \left\langle \boldsymbol{\lambda}^*, \widehat{\boldsymbol{\Psi}} \boldsymbol{v}_{\boldsymbol{\theta}_t, \pi_t} - \boldsymbol{\Psi} \boldsymbol{v}_{\boldsymbol{\theta}_t, \pi_t} \right\rangle + \gamma \left\langle \boldsymbol{\lambda}_t, \widehat{\boldsymbol{\Psi}} \boldsymbol{v}_{\boldsymbol{\theta}_t, \pi_t} - \boldsymbol{\Psi} \boldsymbol{v}_{\boldsymbol{\theta}_t, \pi_t} \right\rangle$$

$$= \left\langle \boldsymbol{\lambda}^* - \boldsymbol{\lambda}_t, \boldsymbol{\omega} + \gamma \widehat{\boldsymbol{\Psi}} \boldsymbol{v}_{\boldsymbol{\theta}_t, \pi_t} - \boldsymbol{\theta}_t \right\rangle - \gamma \left\langle \boldsymbol{\lambda}^*, \widehat{\boldsymbol{\Psi}} \boldsymbol{v}_{\boldsymbol{\theta}_t, \pi_t} - \boldsymbol{\Psi} \boldsymbol{v}_{\boldsymbol{\theta}_t, \pi_t} \right\rangle + \gamma \left\langle \boldsymbol{\lambda}_t, \widehat{\boldsymbol{\Psi}} \boldsymbol{v}_{\boldsymbol{\theta}_t, \pi_t} - \boldsymbol{\Psi} \boldsymbol{v}_{\boldsymbol{\theta}_t, \pi_t} \right\rangle,$$

499 Notice that the last equality follows directly from definition of $\widehat{f}$. Along these lines, we can also
500 express the last set of terms in Equation (11) as follows:

$$f(\boldsymbol{\lambda}_t, \pi_t; \boldsymbol{\theta}_t) - f(\boldsymbol{\lambda}_t, \pi_t; \boldsymbol{\theta}_t^*)$$

$$= \widehat{f}(\boldsymbol{\lambda}_t, \pi_t; \boldsymbol{\theta}_t) - \widehat{f}(\boldsymbol{\lambda}_t, \pi_t; \boldsymbol{\theta}_t^*) - \gamma \left\langle \boldsymbol{\lambda}_t, \widehat{\boldsymbol{\Psi}} \boldsymbol{v}_{\boldsymbol{\theta}_t, \pi_t} - \boldsymbol{\Psi} \boldsymbol{v}_{\boldsymbol{\theta}_t, \pi_t} \right\rangle + \gamma \left\langle \boldsymbol{\lambda}_t, \widehat{\boldsymbol{\Psi}} \boldsymbol{v}_{\boldsymbol{\theta}_t, \pi_t} - \boldsymbol{\Psi} \boldsymbol{v}_{\boldsymbol{\theta}_t^*, \pi_t} \right\rangle$$

$$= \langle \boldsymbol{\theta}_t - \boldsymbol{\theta}_t^*, \boldsymbol{\Phi}^\mathsf{T} \widehat{\boldsymbol{\mu}}_{\boldsymbol{\lambda}_t, \pi_t} - \boldsymbol{\lambda}_t \rangle - \gamma \left\langle \boldsymbol{\lambda}_t, \widehat{\boldsymbol{\Psi}} \boldsymbol{v}_{\boldsymbol{\theta}_t, \pi_t} - \boldsymbol{\Psi} \boldsymbol{v}_{\boldsymbol{\theta}_t, \pi_t} \right\rangle + \gamma \left\langle \boldsymbol{\lambda}_t, \widehat{\boldsymbol{\Psi}} \boldsymbol{v}_{\boldsymbol{\theta}_t, \pi_t} - \boldsymbol{\Psi} \boldsymbol{v}_{\boldsymbol{\theta}_t^*, \pi_t} \right\rangle,$$

501 Plugging the above derivations in the dynamic duality gap, we have that

$$\mathfrak{G}_T(\boldsymbol{\lambda}^*, \pi^*, \boldsymbol{\theta}_{1:T}^*) = \frac{1}{T} \sum_{t=1}^{T} \sum_x \nu^*(x) \sum_a (\pi^*(a|x) - \pi_t(a|x)) q_t(x, a)$$

$$+ \frac{1}{T} \sum_{t=1}^{T} \left\langle \boldsymbol{\lambda}^* - \boldsymbol{\lambda}_t, \boldsymbol{\omega} + \gamma \widehat{\boldsymbol{\Psi}} \boldsymbol{v}_{\boldsymbol{\theta}_t, \pi_t} - \boldsymbol{\theta}_t \right\rangle$$

$$+ \frac{1}{T} \sum_{t=1}^{T} \langle \boldsymbol{\theta}_t - \boldsymbol{\theta}_t^*, \boldsymbol{\Phi}^\mathsf{T} \widehat{\boldsymbol{\mu}}_{\boldsymbol{\lambda}_t, \pi_t} - \boldsymbol{\lambda}_t \rangle$$

$$+ \frac{\gamma}{T} \sum_{t=1}^{T} \left\langle \boldsymbol{\lambda}^*, \boldsymbol{\Psi} \boldsymbol{v}_{\boldsymbol{\theta}_t, \pi_t} - \widehat{\boldsymbol{\Psi}} \boldsymbol{v}_{\boldsymbol{\theta}_t, \pi_t} \right\rangle + \frac{\gamma}{T} \sum_{t=1}^{T} \left\langle \boldsymbol{\lambda}_t, \widehat{\boldsymbol{\Psi}} \boldsymbol{v}_{\boldsymbol{\theta}_t^*, \pi_t} - \boldsymbol{\Psi} \boldsymbol{v}_{\boldsymbol{\theta}_t^*, \pi_t} \right\rangle.$$

502 This matches the claim of the lemma, thus completing the proof. $\qquad\square$

## B.2 Bounding the Regret Terms

In this section we provide the proofs of the our claims made in the main text about the regret of each player—precisely, Lemmas 4.3–4.5.

### B.2.1 Proof of Lemma 4.3

Consider the regret of the $\pi$-player introduced in the main text as,

$$\mathfrak{R}_T\left(\pi^*\right) = \sum_{t=1}^{T} \sum_{x} \nu^*(x) \sum_{a} \left(\pi^*(a|x) - \pi_t(a|x)\right) q_t(x, a)$$

$$\overset{(a)}{\leq} \frac{\sum_x \nu^*(x)\mathcal{D}_{\mathrm{KL}}\left(\pi^*\left(\cdot|x\right)\|\pi_1\left(\cdot|x\right)\right)}{\alpha} + \frac{\alpha T R^2 D_{\boldsymbol{\theta}}^2}{2}$$

$$\overset{(b)}{\leq} \frac{\log A}{\alpha} + \frac{\alpha T R^2 D_{\boldsymbol{\theta}}^2}{2}.$$

We have used $(a)$ the standard Mirror descent analysis of softmax policy iterates recalled in Lemma E.1 for completeness, and $(b)$ the fact that $\pi_1$ is a uniform policy and $\boldsymbol{\nu}^* \in \Delta_{\mathcal{X}}$. Dividing the above expression by $T$ completes the proof. $\qquad\square$

### B.2.2 Proof of Lemma 4.4

Recall the total regret of the $\boldsymbol{\lambda}$-player against any fixed comparator $\boldsymbol{\lambda}^* \in \mathbb{R}^d$ is given as

$$\mathfrak{R}_T\left(\boldsymbol{\lambda}^*\right) = \sum_{t=1}^{T} \langle \boldsymbol{\lambda}^* - \boldsymbol{\lambda}_t, \boldsymbol{\omega} + \gamma \widehat{\boldsymbol{\Psi}} \boldsymbol{v}_{\boldsymbol{\theta}_t, \pi_t} - \boldsymbol{\theta}_t \rangle.$$

Since the feature-occupancy updates of Algorithm 1 simply implements a version of the composite-objective mirror descent scheme due to Duchi et al. [2010] we apply the standard analysis of this method (recalled as Lemma C.1 in Appendix C) to bound the instantaneous regret as

$$\langle \boldsymbol{\lambda}^* - \boldsymbol{\lambda}_t, \boldsymbol{\omega} + \gamma \widehat{\boldsymbol{\Psi}} \boldsymbol{v}_{\boldsymbol{\theta}_t, \pi_t} - \boldsymbol{\theta}_t \rangle$$

$$\leq \frac{\|\boldsymbol{\lambda}_t - \boldsymbol{\lambda}^*\|_{\boldsymbol{\Lambda}_n^{-1}}^2 - \|\boldsymbol{\lambda}_{t+1} - \boldsymbol{\lambda}^*\|_{\boldsymbol{\Lambda}_n^{-1}}^2}{2\eta} + \frac{\eta}{2} \|\boldsymbol{\Lambda}_n \boldsymbol{g}_{\boldsymbol{\lambda}}(t)\|_{\boldsymbol{\Lambda}_n^{-1}}^2 + \frac{\varrho}{2} \|\boldsymbol{\lambda}^*\|_{\boldsymbol{\Lambda}_n^{-1}}^2 - \frac{\varrho}{2} \|\boldsymbol{\lambda}_{t+1}\|_{\boldsymbol{\Lambda}_n^{-1}}^2.$$

Then, taking the sum for $t = 1, \cdots, T$, evaluating the telescoping sums and upper-bounding some negative terms by zero yields the expression

$$\sum_{t=1}^{T} \langle \boldsymbol{\lambda}^* - \boldsymbol{\lambda}_t, \boldsymbol{\omega} + \gamma \widehat{\boldsymbol{\Psi}} \boldsymbol{v}_{\boldsymbol{\theta}_t, \pi_t} - \boldsymbol{\theta}_t \rangle$$

$$\leq \frac{\|\boldsymbol{\lambda}_1 - \boldsymbol{\lambda}^*\|_{\boldsymbol{\Lambda}_n^{-1}}^2}{2\eta} + \frac{\eta}{2} \sum_{t=1}^{T} \|\boldsymbol{\Lambda}_n \boldsymbol{g}_{\boldsymbol{\lambda}}(t)\|_{\boldsymbol{\Lambda}_n^{-1}}^2 + \frac{\varrho T}{2} \|\boldsymbol{\lambda}^*\|_{\boldsymbol{\Lambda}_n^{-1}}^2 - \frac{\varrho}{2} \sum_{t=1}^{T} \|\boldsymbol{\lambda}_t\|_{\boldsymbol{\Lambda}_n^{-1}}^2 + \frac{\varrho}{2} \|\boldsymbol{\lambda}_1\|_{\boldsymbol{\Lambda}_n^{-1}}^2$$

$$= \left(\frac{1}{2\eta} + \frac{\varrho T}{2}\right) \|\boldsymbol{\lambda}^*\|_{\boldsymbol{\Lambda}_n^{-1}}^2 + \frac{\eta}{2} \sum_{t=1}^{T} \|\boldsymbol{\Lambda}_n \boldsymbol{g}_{\boldsymbol{\lambda}}(t)\|_{\boldsymbol{\Lambda}_n^{-1}}^2 - \frac{\varrho}{2} \sum_{t=1}^{T} \|\boldsymbol{\lambda}_t\|_{\boldsymbol{\Lambda}_n^{-1}}^2.$$

In the equality, we have used that $\boldsymbol{\lambda}_1 = \boldsymbol{0}$. Dividing the resulting term by $T$ gives the following bound on the average regret:

$$\frac{1}{T}\mathfrak{R}_T\left(\boldsymbol{\lambda}^*\right) \leq \left(\frac{1}{2T\eta} + \frac{\varrho}{2}\right) \|\boldsymbol{\lambda}^*\|_{\boldsymbol{\Lambda}_n^{-1}}^2 + \frac{\eta}{2T} \sum_{t=1}^{T} \|\boldsymbol{\Lambda}_n \boldsymbol{g}_{\boldsymbol{\lambda}}(t)\|_{\boldsymbol{\Lambda}_n^{-1}}^2 - \frac{\varrho}{2T} \sum_{t=1}^{T} \|\boldsymbol{\lambda}_t\|_{\boldsymbol{\Lambda}_n^{-1}}^2.$$

The proof is completed by applying Lemma C.2 to bound the norm of the gradients and plugging the result into the bound above. $\qquad\square$

 **B.2.3   Proof of Lemma 4.5**

For the regret of the $\boldsymbol{\theta}$-player, first note that for any policy $\pi$ with corresponding state-action value function weights $\boldsymbol{\theta}^\pi = \boldsymbol{\omega} + \gamma \boldsymbol{\Psi} \boldsymbol{v}^\pi$, we have

$$\|\boldsymbol{\theta}^\pi\|_2 = \|\boldsymbol{\omega} + \gamma \boldsymbol{\Psi} \boldsymbol{v}^\pi\|_2 \le \|\boldsymbol{\omega}\|_2 + \gamma \|\boldsymbol{\Psi} \boldsymbol{v}^\pi\|_2 \le \sqrt{d} + \frac{\gamma\sqrt{d}}{(1-\gamma)} = \frac{\sqrt{d}}{(1-\gamma)},$$

where we have used the triangle inequality in the second line. The last inequality uses Definition 2.1 and the fact that $\|\boldsymbol{v}^\pi\|_\infty \le \frac{1}{(1-\gamma)}$ since the rewards are bounded in $[0,1]$. Thanks to this bound, we can ensure that $\boldsymbol{\theta}_t^* = \boldsymbol{\theta}^{\pi_t} \in \mathbb{B}_d(D_{\boldsymbol{\theta}})$ holds with the choice $D_{\boldsymbol{\theta}} = \sqrt{d}/(1-\gamma)$ as required by the lemma. Therefore, by construction of value-parameter updates in Algorithm 1, we have

$$\langle \boldsymbol{\theta}_t - \boldsymbol{\theta}_t^*, \boldsymbol{\Phi}^\top \widehat{\boldsymbol{\mu}}_{\boldsymbol{\lambda}_t, \pi_t} - \boldsymbol{\lambda}_t \rangle \le 0 \qquad \text{for } t = 1, \cdots, T.$$

This concludes the proof. $\qquad\qquad\square$

## B.3   Bounding the gap-estimation error

In this section, we provide the proof of Lemma 4.6 which bounds the gap-estimation error defined for an arbitrary comparator sequence $(\boldsymbol{\lambda}^*, \pi_t, \boldsymbol{\theta}_t^*) \in \mathbb{R}^d \times \Pi(D_\pi) \times \mathbb{B}_d(D_{\boldsymbol{\theta}})$ for $t = 1, \ldots, T$ as,

$$\text{err}_{\widehat{\boldsymbol{\Psi}}} = \sum_{t=1}^T \langle \boldsymbol{\lambda}^*, (\boldsymbol{\Psi} - \widehat{\boldsymbol{\Psi}}) \boldsymbol{v}_{\boldsymbol{\theta}_t, \pi_t} \rangle + \sum_{t=1}^T \langle \boldsymbol{\lambda}_t, (\widehat{\boldsymbol{\Psi}} - \boldsymbol{\Psi}) \boldsymbol{v}_{\boldsymbol{\theta}_t^*, \pi_t} \rangle.$$

We control the above term with the now-classic techniques developed by Jin et al. [2020] for bounding model-estimation errors for linear MDPs. These results also make heavy use of self-normalized tail inequalities as popularized by Abbasi-Yadkori et al. [2011] (see also Lattimore and Szepesvári, 2020). To make this clear, we first note that, for any $\boldsymbol{\lambda} \in \mathbb{R}^d$, $\boldsymbol{v} \in \mathbb{R}^X$, and $\xi > 0$,

$$\left\langle \boldsymbol{\lambda}, \left(\widehat{\boldsymbol{\Psi}} - \boldsymbol{\Psi}\right) \boldsymbol{v} \right\rangle \overset{(a)}{\le} \|\boldsymbol{\lambda}\|_{\boldsymbol{\Lambda}_n^{-1}} \left\|\boldsymbol{\Lambda}_n \left(\widehat{\boldsymbol{\Psi}} - \boldsymbol{\Psi}\right) \boldsymbol{v}\right\|_{\boldsymbol{\Lambda}_n^{-1}} \overset{(b)}{\le} \frac{\|\boldsymbol{\lambda}\|_{\boldsymbol{\Lambda}_n^{-1}}^2}{2T\xi} + \frac{T\xi}{2} \left\|\boldsymbol{\Lambda}_n \left(\widehat{\boldsymbol{\Psi}} - \boldsymbol{\Psi}\right) \boldsymbol{v}\right\|_{\boldsymbol{\Lambda}_n^{-1}}^2.$$

Here, we have first used $(a)$ the Cauchy–Schwarz inequality, and $(b)$ the inequality of arithmetic and geometric means. Using this expression, we can upper-bound the gap estimation error as

$$
\begin{aligned}
\text{err}_{\widehat{\boldsymbol{\Psi}}} \le{} & \frac{\|\boldsymbol{\lambda}^*\|_{\boldsymbol{\Lambda}_n^{-1}}^2}{2\xi} + \sum_{t=1}^T \frac{\|\boldsymbol{\lambda}_t\|_{\boldsymbol{\Lambda}_n^{-1}}^2}{2T\xi} \\
& + \frac{T\xi}{2} \sum_{t=1}^T \left\|\boldsymbol{\Lambda}_n \left(\widehat{\boldsymbol{\Psi}} - \boldsymbol{\Psi}\right) \boldsymbol{v}_{\boldsymbol{\theta}_t, \pi_t}\right\|_{\boldsymbol{\Lambda}_n^{-1}}^2 + \frac{T\xi}{2} \sum_{t=1}^T \left\|\boldsymbol{\Lambda}_n \left(\widehat{\boldsymbol{\Psi}} - \boldsymbol{\Psi}\right) \boldsymbol{v}^{\pi_t}\right\|_{\boldsymbol{\Lambda}_n^{-1}}^2. \quad (12)
\end{aligned}
$$

To control the last two terms in the bound, we employ two main lemmas stated below.

**Lemma B.1.** *Let $\boldsymbol{v} \in [-B, B]^X$. With probability at least $1 - \delta$, we have that:*

$$\left\|\boldsymbol{\Lambda}_n \left(\widehat{\boldsymbol{\Psi}} - \boldsymbol{\Psi}\right) \boldsymbol{v}\right\|_{\boldsymbol{\Lambda}_n^{-1}} \le \frac{2B}{\sqrt{n}} \sqrt{d \log\left(1 + \frac{R^2}{d\beta}\right) + 2\log\frac{1}{\delta}} + B\sqrt{d\beta}.$$

**Lemma B.2.** *Consider the function class,*

$$\mathcal{V} = \left\{ \boldsymbol{v}_{\pi, \boldsymbol{\theta}} : \mathcal{X} \to [-RD_{\boldsymbol{\theta}}, RD_{\boldsymbol{\theta}}] \Big| \pi \in \Pi(D_\pi), \boldsymbol{\theta} \in \mathbb{B}_d(D_{\boldsymbol{\theta}}) \right\},$$

*Let $D_\pi = \alpha T D_{\boldsymbol{\theta}}$ so that $\boldsymbol{v}_{\boldsymbol{\theta}_t, \pi_t} \in \mathcal{V}$. For any $\epsilon \in (0, 1)$, with probability at least $1 - \delta$,*

$$
\begin{aligned}
& \left\|\boldsymbol{\Lambda}_n \left(\widehat{\boldsymbol{\Psi}} - \boldsymbol{\Psi}\right) \boldsymbol{v}_{\boldsymbol{\theta}_t, \pi_t}\right\|_{\boldsymbol{\Lambda}_n^{-1}} \\
& \le \frac{2RD_{\boldsymbol{\theta}}}{\sqrt{n}} \sqrt{d \log\left(1 + \frac{R^2}{d\beta}\right) + 4d \log\left(1 + \frac{4\alpha T R^2 D_{\boldsymbol{\theta}}^2}{\epsilon}\right) + 2\log\frac{1}{\delta}} \\
& \quad + RD_{\boldsymbol{\theta}}\sqrt{d\beta} + \left(\sqrt{\beta} + 1\right) \epsilon \sqrt{d}.
\end{aligned}
$$

543 The rather tedious but otherwise standard proofs of the above lemmas are given in Appendix D. Now,
544 taking into account the fact that for $D_{\boldsymbol{\theta}}$ large enough $\boldsymbol{v}^{\pi_t} \in \mathcal{V}$ yields Corollary B.3 below.

545 **Corollary B.3.** *In the linear MDP setting described in Definition 2.1, notice that $\boldsymbol{v}^{\pi_t} = \boldsymbol{v}_{\boldsymbol{\theta}^{\pi_t}, \pi_t}$ with*
546 $\boldsymbol{\theta}^{\pi_t} = \boldsymbol{\omega} + \gamma \boldsymbol{\Psi} \boldsymbol{v}_{\boldsymbol{\theta}^{\pi_t}, \pi_t}$. *Furthermore, with $RD_{\boldsymbol{\theta}} = R\sqrt{d}/(1-\gamma) \geq \|\boldsymbol{v}^{\pi_t}\|_{\infty}$ and $D_{\pi} = \alpha T D_{\boldsymbol{\theta}}$ we*
547 *have that $\boldsymbol{v}^{\pi_t} \in \mathcal{V}$. Therefore, for all $\epsilon > 0$ with probability at least $1 - \delta$, the following holds:*

$$\left\| \boldsymbol{\Lambda}_n \left( \widehat{\boldsymbol{\Psi}} - \boldsymbol{\Psi} \right) \boldsymbol{v}^{\pi_t} \right\|_{\boldsymbol{\Lambda}_n^{-1}}$$
$$\leq \frac{2\sqrt{d}}{\sqrt{n}\,(1-\gamma)} \sqrt{d \log \left( 1 + \frac{R^2}{d\beta} \right) + 4d \log \left( 1 + \frac{4\alpha T R^2 d}{\epsilon\,(1-\gamma)^2} \right) + 2 \log \frac{1}{\delta}}$$
$$+ \frac{d\sqrt{\beta}}{(1-\gamma)} + \left( \sqrt{\beta} + 1 \right) \epsilon \sqrt{d}.$$

548 In the following, we apply these results to bound the last two terms in the right-hand side of Equa-
549 tion (12). Precisely, using $D_{\boldsymbol{\theta}} = \sqrt{d}/(1-\gamma)$, $\alpha = \sqrt{2 \log A / R^2 D_{\boldsymbol{\theta}}^2 T} = \sqrt{2\,(1-\gamma)^2 \log A / R^2 dT}$
550 (which follows from optimizing the regret of the $\pi$-player in Lemma 4.3), as well as $\epsilon =$
551 $4\alpha R^2 d / (1-\gamma)^2 = \frac{\sqrt{32 R^2 d \log A}}{(1-\gamma)\sqrt{T}}$ and $\beta = R^2/dT$ we have that in any round $t$, with probabil-
552 ity at least $1 - \delta$,

$$\left\| \boldsymbol{\Lambda}_n \left( \widehat{\boldsymbol{\Psi}} - \boldsymbol{\Psi} \right) \boldsymbol{v}_{\boldsymbol{\theta}_t, \pi_t} \right\|_{\boldsymbol{\Lambda}_n^{-1}} \leq \sqrt{\frac{20 d^2 \log\,(2T/\delta)}{n\,(1-\gamma)^2}} + \sqrt{\frac{R^2 d}{T\,(1-\gamma)^2}} + \sqrt{\frac{R^4 d \log A}{T^2}} + \sqrt{\frac{32 R^2 d^2 \log A}{T\,(1-\gamma)^2}}.$$

553 Likewise,

$$\left\| \boldsymbol{\Lambda}_n \left( \widehat{\boldsymbol{\Psi}} - \boldsymbol{\Psi} \right) \boldsymbol{v}^{\pi_t} \right\|_{\boldsymbol{\Lambda}_n^{-1}} \leq \sqrt{\frac{20 d^2 \log\,(2T/\delta)}{n\,(1-\gamma)^2}} + \sqrt{\frac{R^2 d}{T\,(1-\gamma)^2}} + \sqrt{\frac{R^4 d \log A}{T^2}} + \sqrt{\frac{32 R^2 d^2 \log A}{T\,(1-\gamma)^2}}.$$

554 Then plugging in the above bounds in Equation (12) with $T \geq \frac{2R^2 n \log A}{\log(1/\delta)}$, it follows that for
555 $D_{\widehat{\boldsymbol{\Psi}}} = \sqrt{\frac{320 d^2 \log(2T/\delta)}{n(1-\gamma)^2}}$,

$$\mathrm{err}_{\widehat{\boldsymbol{\Psi}}} \leq \frac{\|\boldsymbol{\lambda}^*\|_{\boldsymbol{\Lambda}_n^{-1}}^2}{2\xi} + \sum_{t=1}^{T} \frac{\|\boldsymbol{\lambda}_t\|_{\boldsymbol{\Lambda}_n^{-1}}^2}{2T\xi} + T^2 \xi D_{\widehat{\boldsymbol{\Psi}}}^2,$$

556 with probability at least $1 - \delta$, thus proving the claim. $\qquad \square$

## B.4 Full proof of Theorem 3.1

558 To control the expected suboptimality of the output policy $\pi_J$ of Algorithm 1, we study the repective
559 regret and gap-estimation error at the selected comparator points. Precisely, combining Lemma 4.1
560 and 4.2 when $(\boldsymbol{\lambda}^*, \pi^*, \boldsymbol{\theta}_{1:T}^*) = (\boldsymbol{\Phi}^{\mathsf{T}} \boldsymbol{\mu}^{\pi^*}, \pi^*, \boldsymbol{\theta}^{\pi_t}) \in \mathbb{R}^d \times \Pi\,(D_{\pi}) \times \mathbb{B}_d(D_{\boldsymbol{\theta}})$, we have that,

$$\mathbb{E}_J \left[ \left\langle \boldsymbol{\mu}^{\pi^*} - \boldsymbol{\mu}^{\pi_J}, \boldsymbol{r} \right\rangle \right] = \frac{1}{T} \mathfrak{R}_T\,(\pi^*) + \frac{1}{T} \mathfrak{R}_T\left(\boldsymbol{\lambda}^{\pi^*}\right) + \frac{1}{T} \mathfrak{R}_T\,(\boldsymbol{\theta}_{1:T}^*) + \frac{\gamma}{T} \mathrm{err}_{\widehat{\boldsymbol{\Psi}}}. \qquad (13)$$

561 where,

$$\mathfrak{R}_T\,(\pi^*) = \sum_{t=1}^{T} \sum_x \nu^*(x) \sum_a \left( \pi^*(a|x) - \pi_t(a|x) \right) q_t(x, a),$$

$$\mathfrak{R}_T\left(\boldsymbol{\lambda}^{\pi^*}\right) = \sum_{t=1}^{T} \langle \boldsymbol{\lambda}^{\pi^*} - \boldsymbol{\lambda}_t, \boldsymbol{\omega} + \gamma \widehat{\boldsymbol{\Psi}} \boldsymbol{v}_{\boldsymbol{\theta}_t, \pi_t} - \boldsymbol{\theta}_t \rangle,$$

$$\mathfrak{R}_T\,(\boldsymbol{\theta}_{1:T}^*) = \sum_{t=1}^{T} \langle \boldsymbol{\theta}_t - \boldsymbol{\theta}_t^{\pi_t}, \boldsymbol{\Phi}^{\mathsf{T}} \widehat{\boldsymbol{\mu}}_{\boldsymbol{\lambda}_t, \pi_t} - \boldsymbol{\lambda}_t \rangle$$

$$\mathrm{err}_{\widehat{\boldsymbol{\Psi}}} = \sum_{t=1}^{T} \langle \boldsymbol{\lambda}^{\pi^*}, \left( \boldsymbol{\Psi} - \widehat{\boldsymbol{\Psi}} \right) \boldsymbol{v}_{\boldsymbol{\theta}_t, \pi_t} \rangle + \sum_{t=1}^{T} \langle \boldsymbol{\lambda}_t, \left( \widehat{\boldsymbol{\Psi}} - \boldsymbol{\Psi} \right) \boldsymbol{v}^{\pi_t} \rangle.$$

Notice that for this choice of $\boldsymbol{\lambda}^*$, by Definition 2.1 $\nu^*(x) = (1 - \gamma)\nu_0(x) + \gamma\langle\boldsymbol{\psi}(x), \boldsymbol{\mu}^*\rangle = \nu^{\pi^*}(x)$ is a valid state occupancy measure. Next, introducing the bounds stated in Lemmas 4.3–4.6 under the required conditions $D_{\boldsymbol{\theta}} = \sqrt{d}/(1 - \gamma)$, $\alpha = \sqrt{2(1 - \gamma)^2 \log A/R^2 dT}$, $D_{\pi} = \alpha T D_{\boldsymbol{\theta}} = \sqrt{2T \log A/R^2}$ and $T \geq \frac{2R^2 n \log A}{\log(1/\delta)}$, as well as $\xi \geq 0$ and $D_{\widehat{\boldsymbol{\Psi}}} = \sqrt{\frac{320 d^2 \log(2T/\delta)}{n(1 - \gamma)^2}}$ yields,

$$
\mathbb{E}_J\left[\left\langle\boldsymbol{\mu}^{\pi^*} - \boldsymbol{\mu}^{\pi_J}, \boldsymbol{r}\right\rangle\right] \leq \sqrt{\frac{2R^2 d \log A}{(1 - \gamma)^2 T}}
$$
$$
+ \left(\frac{1}{2\eta T} + \frac{\varrho}{2}\right)\left\|\boldsymbol{\lambda}^{\pi^*}\right\|_{\boldsymbol{\Lambda}_n^{-1}}^2 + \frac{\eta C}{2} - \frac{\varrho}{2T}\sum_{t=1}^T \|\boldsymbol{\lambda}_t\|_{\boldsymbol{\Lambda}_n^{-1}}^2
$$
$$
+ \frac{\gamma}{2T\xi}\left(\left\|\boldsymbol{\lambda}^{\pi^*}\right\|_{\boldsymbol{\Lambda}_n^{-1}}^2 + \frac{1}{T}\sum_{t=1}^T \|\boldsymbol{\lambda}_t\|_{\boldsymbol{\Lambda}_n^{-1}}^2\right) + \gamma T\xi D_{\widehat{\boldsymbol{\Psi}}}^2,
$$

with probability at least $1 - \delta$, where $C = 6\beta\left(d + D_{\boldsymbol{\theta}}^2\right) + 3d\left(1 + RD_{\boldsymbol{\theta}}\right)^2 + 3\gamma^2 dR^2 D_{\boldsymbol{\theta}}^2$. Rearranging the bound and selecting $\varrho = \gamma/\xi T$ to eliminate the (potentially large) norm of the iterates, we obtain

$$
\mathbb{E}_J\left[\left\langle\boldsymbol{\mu}^{\pi^*} - \boldsymbol{\mu}^{\pi_J}, \boldsymbol{r}\right\rangle\right] \leq \sqrt{\frac{d \log(1/\delta)}{n(1 - \gamma)^2}} + \left(\frac{1}{2\eta T} + \frac{\varrho}{2} + \frac{\gamma}{2\xi T}\right)\left\|\boldsymbol{\lambda}^{\pi^*}\right\|_{\boldsymbol{\Lambda}_n^{-1}}^2 + \frac{\eta C}{2}
$$
$$
+ \left(\frac{\gamma}{\xi T} - \varrho\right)\frac{1}{2T}\sum_{t=1}^T \|\boldsymbol{\lambda}_t\|_{\boldsymbol{\Lambda}_n^{-1}}^2 + \gamma T\xi D_{\widehat{\boldsymbol{\Psi}}}^2
$$
$$
= \sqrt{\frac{d \log(1/\delta)}{n(1 - \gamma)^2}} + \left(\frac{1}{2\eta T} + \frac{\gamma}{\xi T}\right)\left\|\boldsymbol{\lambda}^{\pi^*}\right\|_{\boldsymbol{\Lambda}_n^{-1}}^2 + \frac{\eta C}{2} + \gamma T\xi D_{\widehat{\boldsymbol{\Psi}}}^2.
$$

Furthermore, choosing $\xi = 1/TD_{\widehat{\boldsymbol{\Psi}}}$ i.e $\varrho = \gamma D_{\widehat{\boldsymbol{\Psi}}}$, we further simplify the above bound on the regret in terms of the optimization error arising from the policy and feature occupancy updates as,

$$
\mathbb{E}_J\left[\left\langle\boldsymbol{\mu}^{\pi^*} - \boldsymbol{\mu}^{\pi_J}, \boldsymbol{r}\right\rangle\right] \leq \sqrt{\frac{d \log(1/\delta)}{n(1 - \gamma)^2}} + \frac{1}{2\eta T}\left\|\boldsymbol{\lambda}^{\pi^*}\right\|_{\boldsymbol{\Lambda}_n^{-1}}^2 + \frac{\eta C}{2} + \gamma\left(\left\|\boldsymbol{\lambda}^{\pi^*}\right\|_{\boldsymbol{\Lambda}_n^{-1}}^2 + 1\right)D_{\widehat{\boldsymbol{\Psi}}}.
$$

Moving our attention to our earlier bound on the norm of $\boldsymbol{g}_{\boldsymbol{\lambda}}(t)$,

$$
C = 6\beta\left(d + D_{\boldsymbol{\theta}}^2\right) + 3d\left(1 + RD_{\boldsymbol{\theta}}\right)^2 + 3\gamma^2 dR^2 D_{\boldsymbol{\theta}}^2 \leq \frac{27R^2 d^2}{(1 - \gamma)^2}.
$$

The inequality follows from our earlier choice of $\beta = R^2/dT$ and that $T \geq 1/d^2$. Plugging the values of $C$ and $D_{\widehat{\boldsymbol{\Psi}}}$ in the bound, then choosing $\eta = \sqrt{\frac{(1 - \gamma)^2}{27R^2 d^2 T}}$ and using the condition $T \geq \frac{2R^2 n \log A}{\log(1/\delta)}$, we have that with probability at least $1 - \delta$,

$$
\mathbb{E}_J\left[\left\langle\boldsymbol{\mu}^{\pi^*} - \boldsymbol{\mu}^{\pi_J}, \boldsymbol{r}\right\rangle\right] \leq \sqrt{\frac{d \log(1/\delta)}{n(1 - \gamma)^2}} + \left(\left\|\boldsymbol{\lambda}^{\pi^*}\right\|_{\boldsymbol{\Lambda}_n^{-1}}^2 + 1\right)\sqrt{\frac{27d^2 \log(1/\delta)}{8n \log A(1 - \gamma)^2}}
$$
$$
+ \gamma\left(\left\|\boldsymbol{\lambda}^{\pi^*}\right\|_{\boldsymbol{\Lambda}_n^{-1}}^2 + 1\right)\sqrt{\frac{320 d^2 \log(2T/\delta)}{n(1 - \gamma)^2}}
$$
$$
= \mathcal{O}\left(\frac{\left\|\boldsymbol{\lambda}^{\pi^*}\right\|_{\boldsymbol{\Lambda}_n^{-1}}^2 + 1}{(1 - \gamma)}\sqrt{\frac{d^2 \log(2T/\delta)}{n}}\right).
$$

This completes the proof. $\qquad\qquad\qquad\qquad\qquad\qquad\qquad\qquad\qquad\qquad\qquad\quad\square$

# C   Missing proofs of Section B.2

**Lemma C.1.** *(cf. Lemma 1 of Duchi et al. [2010]) Let* $g_{\boldsymbol{\lambda}}(t) = \boldsymbol{\omega} + \gamma\widehat{\boldsymbol{\Psi}}v_{\boldsymbol{\theta}_t,\pi_t} - \boldsymbol{\theta}_t$ *Given* $\boldsymbol{\lambda}_1 = \mathbf{0}$ *and* $\varrho, \eta > 0$ *and the sequence of iterates* $\{\boldsymbol{\lambda}_t\}_{t=2}^{T}$ *defined for* $t = 1, \cdots, T$ *as:*

$$\boldsymbol{\lambda}_{t+1} = \underset{\boldsymbol{\lambda}\in\mathbb{R}^d}{\arg\min}\Big\{ -\langle\boldsymbol{\lambda}, g_{\boldsymbol{\lambda}}(t)\rangle + \frac{1}{2\eta}\|\boldsymbol{\lambda} - \boldsymbol{\lambda}_t\|_{\boldsymbol{\Lambda}_n^{-1}}^2 + \frac{\varrho}{2}\|\boldsymbol{\lambda}\|_{\boldsymbol{\Lambda}_n^{-1}}^2 \Big\}. \tag{14}$$

*Then, for any* $\boldsymbol{\lambda}^* \in \mathbb{R}^d$,

$$\langle \boldsymbol{\lambda}^* - \boldsymbol{\lambda}_t, \boldsymbol{\omega} + \gamma\widehat{\boldsymbol{\Psi}}v_{\boldsymbol{\theta}_t,\pi_t} - \boldsymbol{\theta}_t\rangle$$

$$\leq \frac{\|\boldsymbol{\lambda}_t - \boldsymbol{\lambda}^*\|_{\boldsymbol{\Lambda}_n^{-1}}^2 - \|\boldsymbol{\lambda}_{t+1} - \boldsymbol{\lambda}^*\|_{\boldsymbol{\Lambda}_n^{-1}}^2}{2\eta} + \frac{\eta}{2}\|\boldsymbol{\Lambda}_n g_{\boldsymbol{\lambda}}(t)\|_{\boldsymbol{\Lambda}_n^{-1}}^2 + \frac{\varrho}{2}\|\boldsymbol{\lambda}^*\|_{\boldsymbol{\Lambda}_n^{-1}}^2 - \frac{\varrho}{2}\|\boldsymbol{\lambda}_{t+1}\|_{\boldsymbol{\Lambda}_n^{-1}}^2.$$

*Proof.* The proof of Lemma C.1 follows directly from the referenced Lemma from Duchi et al. [2010]. Consider,

$$\langle\boldsymbol{\lambda}^* - \boldsymbol{\lambda}_t, g_{\boldsymbol{\lambda}}(t)\rangle + \frac{\varrho}{2}\|\boldsymbol{\lambda}_{t+1}\|_{\boldsymbol{\Lambda}_n^{-1}}^2 - \frac{\varrho}{2}\|\boldsymbol{\lambda}^*\|_{\boldsymbol{\Lambda}_n^{-1}}^2$$

$$= \Big\langle\boldsymbol{\lambda}_{t+1} - \boldsymbol{\lambda}^*, -g_{\boldsymbol{\lambda}}(t) + \frac{1}{\eta}\boldsymbol{\Lambda}_n^{-1}(\boldsymbol{\lambda}_{t+1} - \boldsymbol{\lambda}_t) + \varrho\boldsymbol{\Lambda}_n^{-1}\boldsymbol{\lambda}_{t+1}\Big\rangle + \langle\boldsymbol{\lambda}_{t+1} - \boldsymbol{\lambda}_t, g_{\boldsymbol{\lambda}}(t)\rangle$$

$$\quad - \Big\langle\boldsymbol{\lambda}_{t+1} - \boldsymbol{\lambda}^*, \frac{1}{\eta}\boldsymbol{\Lambda}_n^{-1}(\boldsymbol{\lambda}_{t+1} - \boldsymbol{\lambda}_t) + \varrho\boldsymbol{\Lambda}_n^{-1}\boldsymbol{\lambda}_{t+1}\Big\rangle + \frac{\varrho}{2}\|\boldsymbol{\lambda}_{t+1}\|_{\boldsymbol{\Lambda}_n^{-1}}^2 - \frac{\varrho}{2}\|\boldsymbol{\lambda}^*\|_{\boldsymbol{\Lambda}_n^{-1}}^2$$

$$\overset{(a)}{\leq} \langle\boldsymbol{\lambda}_{t+1} - \boldsymbol{\lambda}_t, g_{\boldsymbol{\lambda}}(t)\rangle - \frac{1}{\eta}\langle\boldsymbol{\lambda}_{t+1} - \boldsymbol{\lambda}^*, \boldsymbol{\Lambda}_n^{-1}(\boldsymbol{\lambda}_{t+1} - \boldsymbol{\lambda}_t)\rangle$$

$$\quad + \varrho\langle\boldsymbol{\lambda}^*, \boldsymbol{\Lambda}_n^{-1}\boldsymbol{\lambda}_{t+1}\rangle - \frac{\varrho}{2}\|\boldsymbol{\lambda}_{t+1}\|_{\boldsymbol{\Lambda}_n^{-1}}^2 - \frac{\varrho}{2}\|\boldsymbol{\lambda}^*\|_{\boldsymbol{\Lambda}_n^{-1}}^2$$

$$\overset{(b)}{\leq} \langle\boldsymbol{\lambda}_{t+1} - \boldsymbol{\lambda}_t, g_{\boldsymbol{\lambda}}(t)\rangle + \frac{1}{\eta}\langle\boldsymbol{\lambda}_{t+1} - \boldsymbol{\lambda}^*, \boldsymbol{\Lambda}_n^{-1}(\boldsymbol{\lambda}_t - \boldsymbol{\lambda}_{t+1})\rangle$$

$$\overset{(c)}{=} \langle\boldsymbol{\lambda}_{t+1} - \boldsymbol{\lambda}_t, g_{\boldsymbol{\lambda}}(t)\rangle - \frac{1}{2\eta}\|\boldsymbol{\lambda}_{t+1} - \boldsymbol{\lambda}_t\|_{\boldsymbol{\Lambda}_n^{-1}}^2 + \frac{1}{2\eta}\Big(\|\boldsymbol{\lambda}^* - \boldsymbol{\lambda}_t\|_{\boldsymbol{\Lambda}_n^{-1}}^2 - \|\boldsymbol{\lambda}^* - \boldsymbol{\lambda}_{t+1}\|_{\boldsymbol{\Lambda}_n^{-1}}^2\Big)$$

$$\leq \frac{1}{\eta}\sup_{\boldsymbol{y}\in\mathbb{R}^d}\Big(\langle\boldsymbol{y}, \eta\boldsymbol{\Lambda}_n^{1/2}g_{\boldsymbol{\lambda}}(t)\rangle - \frac{1}{2}\|\boldsymbol{y}\|_2^2\Big) + \frac{1}{2\eta}\Big(\|\boldsymbol{\lambda}^* - \boldsymbol{\lambda}_t\|_{\boldsymbol{\Lambda}_n^{-1}}^2 - \|\boldsymbol{\lambda}^* - \boldsymbol{\lambda}_{t+1}\|_{\boldsymbol{\Lambda}_n^{-1}}^2\Big)$$

$$\overset{(d)}{=} \frac{\eta}{2}\|\boldsymbol{\Lambda}_n g_{\boldsymbol{\lambda}}(t)\|_{\boldsymbol{\Lambda}_n^{-1}}^2 + \frac{1}{2\eta}\Big(\|\boldsymbol{\lambda}^* - \boldsymbol{\lambda}_t\|_{\boldsymbol{\Lambda}_n^{-1}}^2 - \|\boldsymbol{\lambda}^* - \boldsymbol{\lambda}_{t+1}\|_{\boldsymbol{\Lambda}_n^{-1}}^2\Big)$$

We have used

$(a)$ The first order optimality condition on Equation (14):

For any $\boldsymbol{\lambda} \in \mathbb{R}^d$,

$$\Big\langle\boldsymbol{\lambda}_{t+1} - \boldsymbol{\lambda}, -g_{\boldsymbol{\lambda}}(t) + \frac{1}{\eta}\boldsymbol{\Lambda}_n^{-1}(\boldsymbol{\lambda}_{t+1} - \boldsymbol{\lambda}_t) + \varrho\boldsymbol{\Lambda}_n^{-1}\boldsymbol{\lambda}_{t+1}\Big\rangle \leq 0.$$

$(b)$ The relation:

$$\varrho\langle\boldsymbol{\lambda}^*, \boldsymbol{\Lambda}_n^{-1}\boldsymbol{\lambda}_{t+1}\rangle - \frac{\varrho}{2}\|\boldsymbol{\lambda}_{t+1}\|_{\boldsymbol{\Lambda}_n^{-1}}^2 - \frac{\varrho}{2}\|\boldsymbol{\lambda}^*\|_{\boldsymbol{\Lambda}_n^{-1}}^2 = -\frac{\varrho}{2}\|\boldsymbol{\lambda}_{t+1} - \boldsymbol{\lambda}^*\|_{\boldsymbol{\Lambda}_n^{-1}}^2 \leq 0.$$

$(c)$ By definition of the squared $L^2$-norm for vectors $\boldsymbol{a} = \boldsymbol{\Lambda}_n^{-1/2}(\boldsymbol{\lambda}_{t+1} - \boldsymbol{\lambda}^*)$ and $\boldsymbol{b} = \boldsymbol{\Lambda}_n^{-1/2}(\boldsymbol{\lambda}_t - \boldsymbol{\lambda}_{t+1})$:

$$\langle\boldsymbol{a}, \boldsymbol{b}\rangle = \frac{1}{2}\Big(-\|\boldsymbol{b}\|_2^2 + \|\boldsymbol{a} + \boldsymbol{b}\|_2^2 - \|\boldsymbol{a}\|_2^2\Big).$$

Note that $\boldsymbol{a}$ and $\boldsymbol{b}$ are well defined since $\boldsymbol{\Lambda}_n$ is both symmetric and positive definite.

588        $(d)$ By definition of the Fenchel conjugate of $\frac{1}{2}\left\|\boldsymbol{y}\right\|_2^2$ for $\boldsymbol{y} \in \mathbb{R}^d$.

589     Rearranging the terms and plugging in $\boldsymbol{g}_{\boldsymbol{\lambda}}(t) = \boldsymbol{\omega} + \gamma \widehat{\boldsymbol{\Psi}} \boldsymbol{v}_{\boldsymbol{\theta}_t, \pi_t} - \boldsymbol{\theta}_t$ completes the proof.       $\square$

590     Finally, we will use the following result that bounds the gradient norms appearing in the bound above.

591     **Lemma C.2.** *Under the conditions of the linear MDP setting we have that,*

$$\left\|\boldsymbol{\Lambda}_n \boldsymbol{g}_{\boldsymbol{\lambda}}(t)\right\|_{\boldsymbol{\Lambda}_n^{-1}}^2 \le 6\beta \left(d + D_{\boldsymbol{\theta}}^2\right) + 3d\left(1 + RD_{\boldsymbol{\theta}}\right)^2 + 3\gamma^2 dR^2 D_{\boldsymbol{\theta}}^2.$$

592     *Proof.* Recall that for $t = 1, \cdots, T$ $\boldsymbol{g}_{\boldsymbol{\lambda}}(t) = \boldsymbol{\omega} + \gamma \widehat{\boldsymbol{\Psi}} \boldsymbol{v}_{\boldsymbol{\theta}_t, \pi_t} - \boldsymbol{\theta}_t$. Then,

$$\left\|\boldsymbol{\Lambda}_n \boldsymbol{g}_{\boldsymbol{\lambda}}(t)\right\|_{\boldsymbol{\Lambda}_n^{-1}}^2 = \left\|\boldsymbol{\Lambda}_n \left[\boldsymbol{\omega} + \gamma \widehat{\boldsymbol{\Psi}} \boldsymbol{v}_{\boldsymbol{\theta}_t, \pi_t} - \boldsymbol{\theta}_t\right]\right\|_{\boldsymbol{\Lambda}_n^{-1}}^2$$

$$= \left\|\beta \left(\boldsymbol{\omega} - \boldsymbol{\theta}_t\right) + \frac{1}{n}\sum_{i=1}^n \boldsymbol{\varphi}_i \left(r\left(x_i, a_i\right) - \langle \boldsymbol{\varphi}_i, \boldsymbol{\theta}_t \rangle\right) + \gamma \boldsymbol{\Lambda}_n \widehat{\boldsymbol{\Psi}} \boldsymbol{v}_{\boldsymbol{\theta}_t, \pi_t}\right\|_{\boldsymbol{\Lambda}_n^{-1}}^2$$

$$\le 3 \left\|\beta \left(\boldsymbol{\omega} - \boldsymbol{\theta}_t\right)\right\|_{\boldsymbol{\Lambda}_n^{-1}}^2 + 3 \left\|\frac{1}{n}\sum_{i=1}^n \boldsymbol{\varphi}_i \left(r\left(x_i, a_i\right) - \langle \boldsymbol{\varphi}_i, \boldsymbol{\theta}_t \rangle\right)\right\|_{\boldsymbol{\Lambda}_n^{-1}}^2 + 3\gamma^2 \left\|\boldsymbol{\Lambda}_n \widehat{\boldsymbol{\Psi}} \boldsymbol{v}_{\boldsymbol{\theta}_t, \pi_t}\right\|_{\boldsymbol{\Lambda}_n^{-1}}^2.$$

593     Now to bound each of the three terms, we use that

$$\left\|\beta \left(\boldsymbol{\omega} - \boldsymbol{\theta}_t\right)\right\|_{\boldsymbol{\Lambda}_n^{-1}}^2 \le 2\beta^2 \left\|\boldsymbol{\Lambda}_n^{-1}\right\|_2 \left(d + D_{\boldsymbol{\theta}}^2\right) \le 2\beta \left(d + D_{\boldsymbol{\theta}}^2\right),$$

594     where the first inequality uses the assumption that $\left\|\boldsymbol{\omega}\right\|_2 \le \sqrt{d}$ (cf. Definition 2.1) and $\boldsymbol{\theta}_t \in \mathbb{B}_d(D_{\boldsymbol{\theta}})$.
595     Next, we have that

$$\left\|\frac{1}{n}\sum_{i=1}^n \boldsymbol{\varphi}_i \left(r\left(x_i, a_i\right) - \langle \boldsymbol{\varphi}_i, \boldsymbol{\theta}_t \rangle\right)\right\|_{\boldsymbol{\Lambda}_n^{-1}}^2 \le \frac{1}{n}\sum_{i=1}^n \left\|\boldsymbol{\varphi}_i\right\|_{\boldsymbol{\Lambda}_n^{-1}}^2 \left|r\left(x_i, a_i\right) - \langle \boldsymbol{\varphi}_i, \boldsymbol{\theta}_t \rangle\right|^2$$

$$\le d\left(1 + RD_{\boldsymbol{\theta}}\right)^2.$$

596     The last step follows from the fact that the rewards are bounded in $[0, 1]$, $\left\|\boldsymbol{\varphi}_i\right\| \le R$, $\boldsymbol{\theta}_t \in \mathbb{B}_d(D_{\boldsymbol{\theta}})$
597     and Equation (17). The last remaining term is bounded as

$$\left\|\boldsymbol{\Lambda}_n \widehat{\boldsymbol{\Psi}} \boldsymbol{v}_{\boldsymbol{\theta}_t, \pi_t}\right\|_{\boldsymbol{\Lambda}_n^{-1}}^2 = \left\|\frac{1}{n}\sum_{i=1}^n \boldsymbol{\varphi}_i \boldsymbol{v}_{\boldsymbol{\theta}_t, \pi_t}\left(x_i'\right)\right\|_{\boldsymbol{\Lambda}_n^{-1}}^2$$

$$\le \frac{1}{n}\sum_{i=1}^n \left\|\boldsymbol{\varphi}_i\right\|_{\boldsymbol{\Lambda}_n^{-1}}^2 \left\|\boldsymbol{v}_{\boldsymbol{\theta}_t, \pi_t}\right\|_{\infty}^2 \le dR^2 D_{\boldsymbol{\theta}}^2$$

598     Therefore, we obtain

$$\left\|\boldsymbol{\Lambda}_n \boldsymbol{g}_{\boldsymbol{\lambda}}(t)\right\|_{\boldsymbol{\Lambda}_n^{-1}}^2 \le 6\beta \left(d + D_{\boldsymbol{\theta}}^2\right) + 3d\left(1 + RD_{\boldsymbol{\theta}}\right)^2 + 3\gamma^2 dR^2 D_{\boldsymbol{\theta}}^2$$

599     and this completes the proof.       $\square$

# D  Missing proofs of Section B.3

601  In this section, we prove the lemmas stated in Section B.3.

## D.1  Proof of Lemma B.1

603  By definition of $\boldsymbol{\Lambda}_n$ Section 3 and $\widehat{\boldsymbol{\Psi}}$ in Equation (8), we can write:

$$\boldsymbol{\Lambda}_n \left( \widehat{\boldsymbol{\Psi}} - \boldsymbol{\Psi} \right) \boldsymbol{v} = \boldsymbol{\Lambda}_n \left( \frac{1}{n} \boldsymbol{\Lambda}_n^{-1} \sum_{i=1}^n \boldsymbol{\varphi}_i \boldsymbol{e}_{x_i'}^{\mathsf{T}} \right) \boldsymbol{v} - \left( \beta \boldsymbol{I}_n + \frac{1}{n} \sum_{i=1}^n \boldsymbol{\varphi}_i \boldsymbol{\varphi}_i^{\mathsf{T}} \right) \boldsymbol{\Psi} \boldsymbol{v}$$

$$= \frac{1}{n} \sum_{i=1}^n \boldsymbol{\varphi}_i [v(x_i') - \langle \boldsymbol{p}(\cdot | x_i, a_i), \boldsymbol{v} \rangle] - \beta \boldsymbol{\Psi} \boldsymbol{v}$$

604  In the last equality we used definition 2.1 to write $\boldsymbol{\varphi}_i^{\mathsf{T}} \boldsymbol{\Psi} = \boldsymbol{p}(\cdot | x_i, a_i)^{\mathsf{T}}$. Let $\xi_i = v(x_i') -$
605  $\langle \boldsymbol{p}(\cdot | x_i, a_i), \boldsymbol{v} \rangle$. Then,

$$\left\| \boldsymbol{\Lambda}_n \left( \widehat{\boldsymbol{\Psi}} - \boldsymbol{\Psi} \right) \boldsymbol{v} \right\|_{\boldsymbol{\Lambda}_n^{-1}} \le \left\| \frac{1}{n} \sum_{i=1}^n \boldsymbol{\varphi}_i \xi_i \right\|_{\boldsymbol{\Lambda}_n^{-1}} + \| \beta \boldsymbol{\Psi} \boldsymbol{v} \|_{\boldsymbol{\Lambda}_n^{-1}} .$$

606  We easily control the second term with the relation:

$$\| \beta \boldsymbol{\Psi} \boldsymbol{v} \|_{\boldsymbol{\Lambda}_n^{-1}} \le \beta \left\| \boldsymbol{\Lambda}_n^{-1/2} \right\|_2 \| \boldsymbol{\Psi} \boldsymbol{v} \|_2 \le B \sqrt{d\beta} \tag{15}$$

607  The last inequality follows from the fact that $\left\| \boldsymbol{\Lambda}_n^{-1/2} \right\|_2 \le 1/\sqrt{\beta}$ and by definition 2.1 $\| \boldsymbol{\Psi} \boldsymbol{v} \|_2 \le$
608  $B\sqrt{d}$ for $\boldsymbol{v} \in [-B, B]^X$.

609  Now, to handle the first term, let $D_0 = \emptyset$. We construct a filtration $\mathcal{F}_{i-1} = \mathcal{D}_{i-1} \cup (x_i^0, x_i, a_i, r_i)$
610  for $i = 1, 2, \cdots, n$. Notice that by construction of the dataset $\xi_i$ is a martingale difference sequence
611  (i.e $\mathbb{E}[\xi_i | \mathcal{F}_{i-1}] = 0$) taking values in the range $[-2B, 2B]$. Then, we can directly apply Lemma E.3
612  to obtain a bound on the first term as:

$$\left\| \frac{1}{n} \sum_{i=1}^n \boldsymbol{\varphi}_i \xi_i \right\|_{\boldsymbol{\Lambda}_n^{-1}} = \frac{1}{\sqrt{n}} \sqrt{\left\| \sum_{i=1}^n \boldsymbol{\varphi}_i \xi_i \right\|_{(n\boldsymbol{\Lambda}_n)^{-1}}^2} \le \frac{2B}{\sqrt{n}} \sqrt{2 \log \left( \frac{\det (n\boldsymbol{\Lambda}_n)^{1/2} \det (n\beta I)^{-1/2}}{\delta} \right)}$$

$$\le \frac{2B}{\sqrt{n}} \sqrt{d \log \left( 1 + \frac{R^2}{d\beta} \right) + 2 \log \frac{1}{\delta}}.$$

613  with probability $1 - \delta$. In the last inequality we have used the AM-GM inequality and bound on the
614  feature vectors:

$$\det (n\boldsymbol{\Lambda}_n) \le \left( \frac{\operatorname{tr}(n\boldsymbol{\Lambda}_n)}{d} \right)^d = \left( n\beta + \frac{\operatorname{tr}\left( \sum_{i=1}^n \boldsymbol{\varphi}_i \boldsymbol{\varphi}_i^{\mathsf{T}} \right)}{d} \right)^d \le \left( n\beta + \frac{nR^2}{d} \right)^d.$$

615  Putting everything together, we have that w.p $1 - \delta$,

$$\left\| \boldsymbol{\Lambda}_n \left( \widehat{\boldsymbol{\Psi}} - \boldsymbol{\Psi} \right) \boldsymbol{v} \right\|_{\boldsymbol{\Lambda}_n^{-1}} \le \frac{2B}{\sqrt{n}} \sqrt{d \log \left( 1 + \frac{R^2}{d\beta} \right) + 2 \log \frac{1}{\delta}} + B\sqrt{d\beta}.$$

616  This completes the proof.  $\square$

## D.2  Proof of Lemma B.2

618  Unlike Lemma B.1, we now aim to control the error term $\left\| \boldsymbol{\Lambda}_n \left( \widehat{\boldsymbol{\Psi}} - \boldsymbol{\Psi} \right) \boldsymbol{v} \right\|_{\boldsymbol{\Lambda}_n^{-1}}$ when $\boldsymbol{v}$ is random.

619  Also, notice that with $\pi_1(a|x) = \frac{e^{\langle \boldsymbol{\varphi}(x,a), \boldsymbol{0} \rangle}}{\sum_{a' \in \mathcal{A}} e^{\langle \boldsymbol{\varphi}(x,a'), \boldsymbol{0} \rangle}}$ as the uniform policy, for $t = 1, \cdots, T$ we have
620  that,

$$\pi_{t+1}(a|x) = \frac{\pi_1(a|x) e^{\alpha \langle \boldsymbol{\varphi}(x,a), \sum_{k=1}^t \boldsymbol{\theta}_k \rangle}}{\sum_{a'} \pi_1(a'|x) e^{\alpha \langle \boldsymbol{\varphi}(x,a'), \sum_{k=1}^t \boldsymbol{\theta}_k \rangle}} = \frac{e^{\langle \boldsymbol{\varphi}(x,a), \alpha \sum_{k=0}^t \boldsymbol{\theta}_k \rangle}}{\sum_{a'} e^{\langle \boldsymbol{\varphi}(x,a'), \alpha \sum_{k=0}^t \boldsymbol{\theta}_k \rangle}},$$

621 where $\boldsymbol{\theta}_0 = \mathbf{0}$. Furthermore, since $\{\boldsymbol{\theta}_t\}_{t=1}^T \subset \mathbb{B}_d(D_{\boldsymbol{\theta}})$, for any $t$ $\left\| \alpha \sum_{k=0}^t \boldsymbol{\theta}_k \right\|_2 \leq \alpha T D_{\boldsymbol{\theta}}$. Hence,

622 with $D_\pi = \alpha T D_{\boldsymbol{\theta}}$, $\pi_t \in \Pi(D_\pi)$ and $\boldsymbol{v}_{\theta_t, \pi_t} \in \mathcal{V}$.

623 Therefore, as we have seen in previous works [Jin et al., 2020, Hong and Tewari, 2024], the quantity

624 $\left\| \boldsymbol{\Lambda}_n \left( \widehat{\boldsymbol{\Psi}} - \boldsymbol{\Psi} \right) \boldsymbol{v}_{\theta_t, \pi_t} \right\|_{\boldsymbol{\Lambda}_n^{-1}}$ can be controlled without any dependence on the size of the state space

625 with a *uniform covering* argument over $\mathcal{V}$. Let $C_{\boldsymbol{v}}$ be an $\epsilon$-cover of $\mathcal{V}$. That is, for $\boldsymbol{v}_{\pi_t, \theta_t} \in \mathcal{V}$, there

626 exists $\boldsymbol{v}' \in C_{\boldsymbol{v}}$ such that $\|\boldsymbol{v}_{\pi, \theta_t} - \boldsymbol{v}'\|_\infty \leq \epsilon$. Then, we can write:

$$\left\| \boldsymbol{\Lambda}_n \left( \widehat{\boldsymbol{\Psi}} - \boldsymbol{\Psi} \right) \boldsymbol{v}_{\theta_t, \pi_t} \right\|_{\boldsymbol{\Lambda}_n^{-1}}$$
$$\leq \left\| \boldsymbol{\Lambda}_n \left( \widehat{\boldsymbol{\Psi}} - \boldsymbol{\Psi} \right) \boldsymbol{v}' \right\|_{\boldsymbol{\Lambda}_n^{-1}} + \left\| \boldsymbol{\Lambda}_n \widehat{\boldsymbol{\Psi}} \left( \boldsymbol{v}_{\theta_t, \pi_t} - \boldsymbol{v}' \right) \right\|_{\boldsymbol{\Lambda}_n^{-1}} + \left\| \boldsymbol{\Lambda}_n \boldsymbol{\Psi} \left( \boldsymbol{v}' - \boldsymbol{v}_{\theta_t, \pi_t} \right) \right\|_{\boldsymbol{\Lambda}_n^{-1}} \quad (16)$$

627 Consider the first term in the bound. Note that $\boldsymbol{v}'$ is still random with respect to uncertainty in the

628 learning process. However, due to the structure of $\mathcal{V}$ we know that $C_{\boldsymbol{v}}$ exists and has cardinality

629 $\log |C_{\boldsymbol{v}}| = \mathcal{O}\left( d \log \left( 1 + \frac{4 R D_\pi R D_{\boldsymbol{\theta}}}{\epsilon} \right) \right)$ (see Lemma E.6). Inspired by Lemma B.1, consider the event:

$$\mathcal{E}_{\boldsymbol{v}} = \left\{ \text{exists } \boldsymbol{v} \in C_{\boldsymbol{v}} : \left\| \boldsymbol{\Lambda}_n \left( \widehat{\boldsymbol{\Psi}} - \boldsymbol{\Psi} \right) \boldsymbol{v} \right\|_{\boldsymbol{\Lambda}_n^{-1}} > \frac{2 R D_{\boldsymbol{\theta}}}{\sqrt{n}} \sqrt{d \log \left( 1 + \frac{R^2}{d\beta} \right) + 2 \log \frac{1}{\delta'}} + R D_{\boldsymbol{\theta}} \sqrt{d\beta} \right\}$$

630 Since $C_{\boldsymbol{v}} \subseteq \mathcal{V}$, we know from Lemma B.1 that $\mathbb{P}(\mathcal{E}_{\boldsymbol{v}}) \leq \delta'$. Now, taking the union bound over the

631 cover $C_{\boldsymbol{v}}$ we have that,

$$\mathbb{P}\left( \bigcup_{\boldsymbol{v} \in C_{\boldsymbol{v}}} \mathcal{E}_{\boldsymbol{v}} \right) \leq |C_{\boldsymbol{v}}| \delta'.$$

632 Therefore for any $\boldsymbol{v}' \in C_{\boldsymbol{v}}$ with probability at least $1 - \delta$,

$$\left\| \boldsymbol{\Lambda}_n \left( \widehat{\boldsymbol{\Psi}} - \boldsymbol{\Psi} \right) \boldsymbol{v}' \right\|_{\boldsymbol{\Lambda}_n^{-1}}$$
$$\leq \frac{2 R D_{\boldsymbol{\theta}}}{\sqrt{n}} \sqrt{d \log \left( 1 + \frac{R^2}{d\beta} \right) + 2 \log \frac{|C_{\boldsymbol{v}}|}{\delta}} + R D_{\boldsymbol{\theta}} \sqrt{d\beta}$$
$$\leq \frac{2 R D_{\boldsymbol{\theta}}}{\sqrt{n}} \sqrt{d \log \left( 1 + \frac{R^2}{d\beta} \right) + 4 d \log \left( 1 + \frac{4 R D_\pi R D_{\boldsymbol{\theta}}}{\epsilon} \right) + 2 \log \frac{1}{\delta}} + R D_{\boldsymbol{\theta}} \sqrt{d\beta}$$

633 Now, for the second term in Equation (16) we write,

$$\left\| \boldsymbol{\Lambda}_n \widehat{\boldsymbol{\Psi}} \left( \boldsymbol{v}_{\theta_t, \pi_t} - \boldsymbol{v}' \right) \right\|_{\boldsymbol{\Lambda}_n^{-1}}^2 = \left\| \frac{1}{n} \sum_{i=1}^n \boldsymbol{\varphi}_i \left( \boldsymbol{v}_{\theta_t, \pi_t}(x_i') - \boldsymbol{v}'(x_i') \right) \right\|_{\boldsymbol{\Lambda}_n^{-1}}^2$$
$$\overset{(a)}{\leq} \frac{1}{n} \sum_{i=1}^n \left| \boldsymbol{v}_{\theta_t, \pi_t}(x_i') - \boldsymbol{v}'(x_i') \right|^2 \| \boldsymbol{\varphi}_i \|_{\boldsymbol{\Lambda}_n^{-1}}^2$$
$$\leq \epsilon^2 \frac{1}{n} \sum_{i=1}^n \| \boldsymbol{\varphi}_i \|_{\boldsymbol{\Lambda}_n^{-1}}^2 \overset{(b)}{\leq} \epsilon^2 d.$$

634 We have used $(a)$ Jensen's inequality and $(b)$ since $\boldsymbol{\Lambda}_n \succ 0$, the relation,

$$\frac{1}{n} \sum_{i=1}^n \boldsymbol{\varphi}_i^\top \boldsymbol{\Lambda}_n^{-1} \boldsymbol{\varphi}_i = \frac{1}{n} \sum_{i=1}^n \text{tr}\left( \boldsymbol{\Lambda}_n^{-1} \boldsymbol{\varphi}_i \boldsymbol{\varphi}_i^\top \right) = \text{tr}\left( \boldsymbol{\Lambda}_n^{-1} \frac{1}{n} \sum_{i=1}^n \boldsymbol{\varphi}_i \boldsymbol{\varphi}_i^\top \right) \leq \text{tr}(I) = d. \quad (17)$$

For the last term, notice that:

$$
\begin{aligned}
\left\|\boldsymbol{\Lambda}_n \boldsymbol{\Psi}\left(\boldsymbol{v}'-\boldsymbol{v}_{\theta_t, \pi_t}\right)\right\|_{\boldsymbol{\Lambda}_n^{-1}} &= \left\|\beta \boldsymbol{\Psi}\left(\boldsymbol{v}'-\boldsymbol{v}_{\theta_t, \pi_t}\right)+\frac{1}{n} \sum_{i=1}^n \boldsymbol{\varphi}_i\left[\sum_{x'} p\left(x' \mid x_i, a_i\right)\left(\boldsymbol{v}'\left(x'\right)-\boldsymbol{v}_{\theta_t, \pi_t}\left(x'\right)\right)\right]\right\|_{\boldsymbol{\Lambda}_n^{-1}} \\
&\overset{(a)}{\leq} \epsilon \sqrt{d \beta}+\sqrt{\left\|\frac{1}{n} \sum_{i=1}^n \boldsymbol{\varphi}_i\left[\sum_{x'} p\left(x' \mid x_i, a_i\right)\left(\boldsymbol{v}'\left(x'\right)-\boldsymbol{v}_{\theta_t, \pi_t}\left(x'\right)\right)\right]\right\|_{\boldsymbol{\Lambda}_n^{-1}}^2} \\
&\overset{(b)}{\leq} \epsilon \sqrt{d \beta}+\sqrt{\frac{1}{n} \sum_{i=1}^n\left\|\boldsymbol{v}'-\boldsymbol{v}_{\theta_t, \pi_t}\right\|_{\infty}^2\left\|\boldsymbol{\varphi}_i\right\|_{\boldsymbol{\Lambda}_n^{-1}}^2} \\
&\overset{(c)}{\leq} \epsilon \sqrt{d \beta}+\epsilon \sqrt{d}=\epsilon \sqrt{d}\left(\sqrt{\beta}+1\right) .
\end{aligned}
$$

This follows from $(a)$ Equation (15) since $\boldsymbol{v}=\boldsymbol{v}'-\boldsymbol{v}_{\theta_t, \pi_t} \in[-\epsilon, \epsilon]^X$ and $(b)$ monotonicity of the square root function as well as Jensen's inequality and $(c)$ Equation (17).

Finally, plugging the above results back into Equation (16), we have that with probability at least $1-\delta$,

$$
\begin{aligned}
&\left\|\boldsymbol{\Lambda}_n\left(\widehat{\boldsymbol{\Psi}}-\boldsymbol{\Psi}\right) \boldsymbol{v}_{\theta_t, \pi_t}\right\|_{\boldsymbol{\Lambda}_n^{-1}} \\
&\quad \leq \frac{2 R D_{\boldsymbol{\theta}}}{\sqrt{n}} \sqrt{d \log \left(1+\frac{R^2}{d \beta}\right)+4 d \log \left(1+\frac{4 R D_{\pi} R D_{\boldsymbol{\theta}}}{\epsilon}\right)+2 \log \frac{1}{\delta}} \\
&\quad\quad + R D_{\boldsymbol{\theta}} \sqrt{d \beta}+\left(\sqrt{\beta}+1\right) \epsilon \sqrt{d}
\end{aligned}
$$

The proof of Lemma B.2 is complete. $\qquad\square$

## E  Auxiliary Lemmas

**Lemma E.1.** *Let $q_1, \cdots, q_t$ be a sequence of iterates satisfying $\|q_t\|_\infty \leq RD_{\boldsymbol{\theta}}$ by virtue of definition 2.1 and $\boldsymbol{\theta}_t \in \mathbb{B}_d(D_{\boldsymbol{\theta}})$. Given an initial policy $\pi_1$ and learning rate $\alpha > 0$, and sequence of policies $\{\pi_t\}_{t=2}^T$ defined as:*

$$\pi_{t+1}(a|x) = \frac{\pi_t(a|x)e^{\alpha q_t(x,a)}}{\sum_{a'} \pi_t(a'|x)e^{\alpha q_t(x,a')}},$$

*Then, for any comparator policy $\pi^*$ and $\nu^*$ some state distribution,*

$$\sum_{t=1}^T \sum_x \nu^*(x) \sum_a \left(\pi^*(a|x) - \pi_t(a|x)\right) q_t(x,a) \leq \frac{\sum_x \nu^*(x)\mathcal{D}_{\text{KL}}\left(\pi^*(\cdot|x)\|\pi_1(\cdot|x)\right)}{\alpha} + \frac{\alpha T R^2 D_{\boldsymbol{\theta}}^2}{2}.$$

The proof of the lemma follows from bounding the regret of the $\pi$-player in each state $x$ as

$$\sum_{t=1}^T \sum_a \left(\pi^*(a|x) - \pi_t(a|x)\right) q_t(x,a) \leq \frac{\mathcal{D}_{\text{KL}}\left(\pi^*(\cdot|x)\|\pi_1(\cdot|x)\right)}{\alpha} + \frac{\alpha}{2} \sum_{t=1}^T \|q_t(x,\cdot)\|_\infty^2,$$

via the application of the standard analysis of the exponentially weighted forecaster of Vovk [1990], Littlestone and Warmuth [1994], Freund and Schapire [1997] (see, e.g., Theorem 2.2 in Cesa-Bianchi and Lugosi, 2006), and noting that $\|q_t\|_\infty \leq RD_{\boldsymbol{\theta}}$ for all $t$.

**Lemma E.2.** *Suppose that $\|\boldsymbol{\varphi}(x,a)\|_2 \leq R$ for all $(x,a) \in \mathcal{X} \times \mathcal{A}$. Let $\pi_{\boldsymbol{\theta}}, \pi_{\boldsymbol{\theta}'}$ be softmax policies. Then, for all states $x \in \mathcal{X}$ we have that:*

$$\sum_a |\pi_{\boldsymbol{\theta}}(a|x) - \pi_{\boldsymbol{\theta}'}(a|x)| \leq R \|\boldsymbol{\theta} - \boldsymbol{\theta}'\|_2$$

*holds for any $\boldsymbol{\theta}, \boldsymbol{\theta}' \in \mathbb{R}^d$.*

*Proof.* Recall that,

$$\Pi(D_\pi) = \left\{ \pi_\theta(a|x) = \frac{e^{\langle \boldsymbol{\varphi}(x,a), \boldsymbol{\theta}\rangle}}{\sum_{a'} e^{\langle \boldsymbol{\varphi}(x,a'), \boldsymbol{\theta}\rangle}} \;\middle|\; \boldsymbol{\theta} \in \mathbb{B}_d(D_\pi) \right\}.$$

For $\pi_{\boldsymbol{\theta}}, \pi_{\boldsymbol{\theta}'} \in \Pi(D_\pi)$ using Pinsker's inequality we have that,

$$\|\pi_{\boldsymbol{\theta}}(\cdot|x) - \pi_{\boldsymbol{\theta}'}(\cdot|x)\|_1 \leq \sqrt{2\mathcal{D}_{\text{KL}}\left(\pi_{\boldsymbol{\theta}}(\cdot|x)\|\pi_{\boldsymbol{\theta}'}(\cdot|x)\right)} \qquad \text{for } x \in \mathcal{X}. \tag{18}$$

Furthermore, taking into account the specific structure of the policies, we can write:

$$\begin{aligned}
\mathcal{D}_{\text{KL}}\left(\pi_{\boldsymbol{\theta}}(\cdot|x)\|\pi_{\boldsymbol{\theta}'}(\cdot|x)\right) &= \sum_a \pi_{\boldsymbol{\theta}}(a|x) \log \frac{\pi_{\boldsymbol{\theta}}(a|x)}{\pi_{\boldsymbol{\theta}'}(a|x)} \\
&= -\sum_a \pi_{\boldsymbol{\theta}}(a|x) \langle \boldsymbol{\varphi}(x,a), \boldsymbol{\theta}' - \boldsymbol{\theta}\rangle + \log \frac{\sum_a e^{\langle \boldsymbol{\varphi}(x,a), \boldsymbol{\theta}'\rangle}}{\sum_a e^{\langle \boldsymbol{\varphi}(x,a), \boldsymbol{\theta}\rangle}} \\
&\stackrel{(a)}{=} -\sum_a \pi_{\boldsymbol{\theta}}(a|x) \langle \boldsymbol{\varphi}(x,a), \boldsymbol{\theta}' - \boldsymbol{\theta}\rangle + \log \sum_a \pi_{\boldsymbol{\theta}}(a|x) e^{\langle \boldsymbol{\varphi}(x,a), \boldsymbol{\theta}' - \boldsymbol{\theta}\rangle} \\
&\stackrel{(b)}{=} \frac{R^2 \|\boldsymbol{\theta} - \boldsymbol{\theta}'\|_2^2}{2}
\end{aligned}$$

using that $(a)$ the relation,

$$\log \frac{\sum_a e^{\langle \boldsymbol{\varphi}(x,a), \boldsymbol{\theta}'\rangle}}{\sum_a e^{\langle \boldsymbol{\varphi}(x,a), \boldsymbol{\theta}\rangle}} = \log \sum_a \frac{e^{\langle \boldsymbol{\varphi}(x,a), \boldsymbol{\theta}'\rangle}}{\sum_{a'} e^{\langle \boldsymbol{\varphi}(x,a'), \boldsymbol{\theta}\rangle}} \cdot \frac{e^{\langle \boldsymbol{\varphi}(x,a), \boldsymbol{\theta}\rangle}}{e^{\langle \boldsymbol{\varphi}(x,a), \boldsymbol{\theta}\rangle}} = \log \sum_a \pi_{\boldsymbol{\theta}}(a|x) e^{\langle \boldsymbol{\varphi}(x,a), \boldsymbol{\theta}' - \boldsymbol{\theta}\rangle},$$

and $(b)$ Hoeffding's lemma (cf. Lemma A.1 of Cesa-Bianchi and Lugosi [2006]). The final statement follows from substituting this result in Equation (18). $\qquad\square$

**Lemma E.3.** *(Self-Normalized Bound for Vector-Valued Martingales - Theorem 1 of Abbasi-Yadkori et al. [2011]) Let $\{\mathcal{F}_{i-1}\}_{i=1}^{\infty}$ be a filtration and $\{\xi_i\}_{i=1}^{\infty}$ a real-valued stochastic process such that $\xi_i$ for $i = 1, \cdots$ is zero-mean (i.e $\mathbb{E}[\xi_i | \mathcal{F}_{i-1}] = 0$) and conditionally $s$-subgaussian for $s \geq 0$. That is, for all $b \in \mathbb{R}$,*

$$\mathbb{E}\left[e^{b\xi_i} | \mathcal{F}_{i-1}\right] \leq e^{\frac{b^2 s^2}{2}}.$$

*Also, let $\{\boldsymbol{\varphi}_i\}_{i=1}^{\infty}$ be $\mathcal{F}_{i-1}$-measurable. Then,*

$$\left\|\sum_{i=1}^{n} \boldsymbol{\varphi}_i \xi_i\right\|_{(n\boldsymbol{\Lambda}_n)^{-1}}^2 \leq 2s^2 \log\left[\frac{\det(n\boldsymbol{\Lambda}_n)^{1/2} \det(n\beta I)^{-1/2}}{\delta}\right].$$

**Lemma E.4.** *(e.g. see Chapter 27 of Shalev-Shwartz and Ben-David [2014]) For all $\epsilon > 0$,*

$$\log \mathcal{N}\left(\mathbb{B}_d(r), \|\cdot\|_{\infty}, \epsilon\right) \leq d \log\left(1 + \frac{2r}{\epsilon}\right).$$

**Corollary E.5.** *Under the conditions of Lemma E.2, for all $\epsilon > 0$,*

$$\log \mathcal{N}\left(\Pi(D_\pi), \|\cdot\|_{\infty,1}, \epsilon\right) \leq \log \mathcal{N}\left(\mathbb{B}_d(D_\pi), \|\cdot\|_{\infty}, \frac{\epsilon}{R}\right) \leq d \log\left(1 + \frac{2RD_\pi}{\epsilon}\right).$$

**Lemma E.6.** *Consider the function class,*

$$\mathcal{V} = \left\{\boldsymbol{v}_{\pi,\boldsymbol{\theta}} : \mathcal{X} \to [-RD_{\boldsymbol{\theta}}, RD_{\boldsymbol{\theta}}] \Big| \pi \in \Pi(D_\pi), \boldsymbol{\theta} \in \mathbb{B}_d(D_{\boldsymbol{\theta}})\right\},$$

*we have that:*

$$\mathcal{N}\left(\mathcal{V}, \|\cdot\|_{\infty}, \epsilon\right) \leq \mathcal{N}\left(\Pi(D_\pi), \|\cdot\|_{\infty,1}, \epsilon/2RD_{\boldsymbol{\theta}}\right) \times \mathcal{N}\left(\mathbb{B}_d(D_{\boldsymbol{\theta}}), \|\cdot\|_2, \epsilon/2R\right),$$

*and,*

$$\log \mathcal{N}\left(\mathcal{V}, \|\cdot\|_{\infty}, \epsilon\right) \leq 2d \log\left(1 + \frac{4RD_\pi RD_{\boldsymbol{\theta}}}{\epsilon}\right)$$

*Proof.* Let $C_\pi$ denote the $\epsilon_\pi$-cover of $\Pi(D_\pi)$ with respect to the norm $\|\cdot\|_{\infty,1}$ and $C_{\boldsymbol{\theta}}$ the $\epsilon_{\boldsymbol{\theta}}$-cover of $\mathbb{B}_d(D_{\boldsymbol{\theta}})$ under the $L^2$-norm. For $(\pi, \boldsymbol{\theta}) \in \Pi(D_\pi) \times \mathbb{B}_d(D_{\boldsymbol{\theta}})$ and $(\pi', \boldsymbol{\theta}') \in C_\pi \times C_{\boldsymbol{\theta}}$, it follows that for any state $x \in \mathcal{X}$,

$$\begin{aligned}
|\boldsymbol{v}_{\pi,\boldsymbol{\theta}}(s) - \boldsymbol{v}_{\pi',\boldsymbol{\theta}'}(s)| &= \left|\sum_{a \in \mathcal{A}} \pi(a|x) \langle \boldsymbol{\varphi}(x,a), \boldsymbol{\theta}\rangle - \pi'(a|x) \langle \boldsymbol{\varphi}(x,a), \boldsymbol{\theta}'\rangle\right| \\
&= \left|\sum_{a \in \mathcal{A}} (\pi(a|x) - \pi'(a|x)) \langle \boldsymbol{\varphi}(x,a), \boldsymbol{\theta}\rangle + \sum_{a \in \mathcal{A}} \pi'(a|x) \langle \boldsymbol{\varphi}(x,a), \boldsymbol{\theta} - \boldsymbol{\theta}'\rangle\right| \\
&\leq RD_{\boldsymbol{\theta}} \sum_{a \in \mathcal{A}} |\pi(a|x) - \pi'(a|x)| + R \sum_{a \in \mathcal{A}} \pi'(a|x) \|\boldsymbol{\theta} - \boldsymbol{\theta}'\|_2
\end{aligned}$$

Let $C_{\boldsymbol{v}} = \left\{\boldsymbol{v}_{\pi,\boldsymbol{\theta}} : \mathcal{X} \to [-RD_{\boldsymbol{\theta}}, RD_{\boldsymbol{\theta}}] \Big| \pi \in C_\pi, \boldsymbol{\theta} \in C_{\boldsymbol{\theta}}\right\}$. Then, $C_{\boldsymbol{v}}$ is an $\epsilon$-cover of $\mathcal{V}$ with respect to the $L^\infty$-norm when $\epsilon_\pi = \epsilon/2RD_{\boldsymbol{\theta}}$ and $\epsilon_{\boldsymbol{\theta}} = \epsilon/2R$. Therefore, we can derive a bound on the covering number of $C_{\boldsymbol{v}}$ as:

$$\begin{aligned}
\mathcal{N}\left(\mathcal{V}, \|\cdot\|_{\infty}, \epsilon\right) &\leq \mathcal{N}\left(\Pi(D_\pi), \|\cdot\|_{\infty,1}, \epsilon/2RD_{\boldsymbol{\theta}}\right) \times \mathcal{N}\left(\mathbb{B}_d(D_{\boldsymbol{\theta}}), \|\cdot\|_2, \epsilon/2R\right) \\
&\leq \left(1 + \frac{4RD_\pi RD_{\boldsymbol{\theta}}}{\epsilon}\right)^d \left(1 + \frac{4RD_{\boldsymbol{\theta}}}{\epsilon}\right)^d.
\end{aligned}$$

Hence,

$$\log \mathcal{N}\left(\mathcal{V}, \|\cdot\|_{\infty}, \epsilon\right) \leq 2d \log\left(1 + \frac{4RD_\pi RD_{\boldsymbol{\theta}}}{\epsilon}\right)$$

This completes the proof. $\qquad\square$

