# OpenReview forum: "Offline RL via Feature-Occupancy Gradient Ascent"
_NeurIPS.cc/2024/Conference — Submitted to NeurIPS 2024_

### Official Review · Reviewer_5hYq · 2024-07-09

**Soundness:** 4
**Presentation:** 3
**Contribution:** 3
**Rating:** 6
**Confidence:** 4

**Summary:**

This paper studies the offline policy optimization problem, i.e. to find a policy whose value function is close to the optimal value function using offline samples.  Under the assumption of linear MDP, they proposed a gradient ascent algorithm.

The sample complexity of the algorithm only depends on the feature coverage of the best policy and does not require coverage over any other policies.

**Strengths:**

This paper is well written. The algorithm, theorems and lemmas presented in this paper are all very clear.

The problem studied in this paper is offline policy optimization, which has significant value to the community, both empirically and theoretically.

The algorithm in this paper is simple and computationally tractable.

In contrast to other offline RL paper, this paper does not require that the offline data is sampled i.i.d. or to be admissible, and they can handle arbitrary offline data as long as the data has sufficient coverage to the best policy.

**Weaknesses:**

Compared to Zanette (2021), the algorithm idea is somehow similar. Specifically, both algorithms use the actor-critic update, and the optimization in the algorithm of this paper is similar to the pessimism estimation in Zanette (2021).

The assumption made in this paper is the linear MDP assumption, which is stronger than the assumption in Zanette (2021).

**Questions:**

Can your results extend to the case under linear Q-function and linear Bellman completeness assumptions?

**Limitations:**

Yes. The authors addressed all the limitations listed in the guidelines.

---

> ### Author Rebuttal · Authors · 2024-08-05
>
> Thank you for your positive review of our work. We particularly
> appreciate your relation of our method to the actor-critic style framework.
> Please see our response to your remarks and questions below.
>
> &nbsp;
>
> **Weaknesses**
>
> **Q1.** Compared to Zanette (2021), the algorithm idea is somehow similar.
> Specifically, both algorithms use the actor-critic update, and the optimization
> in the algorithm of this paper is similar to the pessimism estimation in
> Zanette (2021).
>
> **A.** Indeed, our method is somewhat similar in spirit to the actor-critic style method of
> 	Zanette (2021) with the critic handling $\mathbf{\theta}$ and the actor performing a sequence of softmax policy updates.
> 	However, there are some notable differences in how the critic updates are implemented in the two methods.
> 	The most remarkable difference is that our critic updates do not involve any explicit
> 	form of pessimism, which makes our algorithm significantly simpler. To see this,
> 	note that the critic update of Zanette et al. (2011) in Eq.(10) requires solving
> 	a quadratically constrained convex optimization problem, which is arguably more
> 	computationally intense than our critic update (which only requires minimizing a linear
> 	function on a norm ball). Furthermore, their algorithm is specifically developed
> 	for the finite-horizon setting and extending it to the infinite-horizon setting would
> 	run into some well-known computational feasibility issues (as pointed out by Wei et al., 2021).
>       Overall, our method is quite different from regular actor-critic methods in that it places feature occupancies $\lambda$ in the spotlight -- as opposed to treating the value parameters $\theta$ and the policy updates as the main characters as done by actor-critic methods.
>
> **Q2.**  The assumption made in this paper is the linear MDP assumption, which is
> stronger than the assumption in Zanette (2021).
>
> **A** Indeed, the assumptions we make in this paper are stronger than the
> $Q^{\pi}$-realizability and Bellman completeness assumption in Zanette (2021).
> However, we consider the more complex infinite-horizon setting for which there
> are no known sample and computationally efficient (not oracle-efficient)
> methods that work under $Q^{\pi}$-realizability and Bellman completeness.
> Extending our results to continue working under less restrictive assumptions
> on the function approximation is an exciting open question that we had to leave
> open for now.
>
>
>
> &nbsp;
>
> **Questions**
>
> **Q1.** Can your results extend to the case under linear Q-function and linear
> Bellman completeness assumptions?
>
> **A** We doubt that such extension is possible with the current setup. Our
> current
> results heavily exploit the linear MDP assumption to first reduce the number of
> decision variables of the standard primal LP in Eq. 1 and introduce $\mathbf{\lambda}$,
> then design the reduced approximate objective $\hat{f}(\mathbf{\lambda},\pi,\mathbf{\theta})$.
> Extending our results to the case of $Q^{\pi}$-realizability and Bellman
> completeness would require first constructing a separate yet similarly reduced
> objective under these weaker assumptions. While we are unsure of how to
> construct such an objective at the moment, we are
> optimistic that the ideas and optimization techniques developed in the paper
> can be applied to handle offline RL under $Q^{\pi}$-realizability and Linear
> Bellman completeness \emph{for all policies}. As we highlight in Section 5 of
> the paper, this is indeed an interesting future direction of our work.

---

> > ### Comment · Reviewer_5hYq · 2024-08-12
> >
> > Thank you very much for your response. I do not have further questions.

---

### Official Review · Reviewer_J7AJ · 2024-07-15

**Soundness:** 3
**Presentation:** 3
**Contribution:** 3
**Rating:** 7
**Confidence:** 4

**Summary:**

This paper propose a new algorithm for offline reinforcement learning in linear infinite-horizon discounted MDP, which achieves a strong sample complexity under the weakest data coverage assumption. Moreover, their algorithm is easy to implement and computationally efficient. Their algorithm design is based on a reduced version of linear programming formulation of MDP, which approximately transforms the original problem into a unconstrained saddle-point optimization problem.

**Strengths:**

1. The algorithm proposed in this paper has many nice properties: it is computationally efficient, easy to implement, and works under the weakest data coverage assumption.
2. The developed techniques and the observations on transforming the optimization problem is insightful.
3. The paper is also well-written, with clear proof sketch and is well-positioned among related work.

**Weaknesses:**

The paper is pretty notation heavy and a bit hard to follow. I would suggest including a table of notations with descriptions. The choices of notation can also be optimized. For example, I found the mixed use of $D_{\pi}$ and $D_{\theta}$ confusing, and they looks like they are dependent on the value of $\pi$ and $\theta$.

**Questions:**

1. is there a way to output a single policy (not a uniformly sampled policy) that achieves the same sample complexity?
2. I am wondering whether your algorithm can be extended to the stochastic shortest path setting [1, 2], which include finite-horizon MDP and discounted infinite-horizon MDP as special cases.

[1] Yin, M., Chen, W., Wang, M., and Wang, Y.-X. Offline stochastic shortest path: Learning, evaluation and towards optimality
[2] Chen, L., Jain, R., and Luo, H. Improved no-regret algorithms for stochastic shortest path with linear mdp

**Limitations:**

Nothing necessary stands out.

---

> ### Author Rebuttal · Authors · 2024-08-05
>
> Thank you for your detailed review of our work. We appreciate your feedback on our notation and will consider your suggestion in the next draft of our paper. For now, please see our response to your questions below:
>
> &nbsp;
>
> **Q1.** Is there a way to output a single policy (not a uniformly sampled policy)
> that achieves the same sample complexity?
>
> **A.** This is a great question, and the answer is currently unclear. The nature of the output policy allows us the
> 	relate the duality gap (introduced in Section 4) directly to the policy
> 	suboptimality, and importantly the average regret of all the players --
> 	which from existing literature on convex optimization can be easily
> 	controlled. Note that the first part would still be
> 	straightforward if we were to output the last policy iterate for example.
> 	However, it is currently unclear how to prove last iterate convergence of the instances of mirror descent and composite-objective gradient ascent which we use for the $\pi$-player and the $\mathbf{\lambda}$-player respectively (the challenge being that the sequence of $\theta$ variables are adversarial from the perspective of these two aforementioned players).
>
> **Q2.** I am wondering whether your algorithm can be extended to the stochastic
> shortest path setting [1, 2], which include finite-horizon MDP and discounted
> infinite-horizon MDP as special cases.
>
> **A.** Thank you for bringing up this great question! Since our approach is
> based on the LP formulation of optimal control in MDPs, we believe that it should be
> extensible without major hassle to all other problem variants for which optimal solutions
> can be formulated as LPs. These include finite-horizon MDPs, infinite-horizon undiscounted MDPs (under the assumption
> that all policies mix to their stationary distributions), and, yes, stochastic shortest-path problems. The analysis of
> the resulting method should be more or less straightforward if one assumes that all policies are proper (in the sense
> that they reach the terminal state after a finite number of steps on expectation). Without such assumption, however, it
> is unclear if our approach could work --- this is definitely an interesting question for future work.

---

### Official Review · Reviewer_5zRN · 2024-07-21

**Soundness:** 3
**Presentation:** 3
**Contribution:** 2
**Rating:** 3
**Confidence:** 3

**Summary:**

This paper provides an approach for solving Offline RL problems in domains where the underlying MDP problem has reward and transition models that are linearly realisable under a known feature map. The paper is well presented.

**Strengths:**

1. Some of the tricks employed in the approach are quite interesting.
2. The analytical results are good.

**Weaknesses:**

1. This work is only for MDPs where rewards and transitions are linear. Many of the moderately interesting problems are not linearly realisable, so it is quite important that authors provided a detailed discussion on how it can be addressed.
2. I am not entirely confident about this, so will wait for inputs from authors. The approach seems to be built based on works by Hong and Tiwari, and stabilisation trick [Neu and okolo, Jacobson et al.]. I was not sure on the key significant contributions of this paper on top of those works.
3. For me the biggest concern is that there are no experimental results. How would such an approach work for an MDP where the transition and reward models are not linear? Also, the approach is still approximate (given the bound), so would have been important to show the real results.

**Questions:**

Apart from the questions mentioned above in weaknesses, can the authors also provide answers to the following questions:
1. Theorem 3.1 is mentioned as the most important results. However, I am not clear on why it is a significant result? Why is this bound good?
2. Also the feature coverage ratio provided by this algorithm, why is it significant? And do others also achieve this ratio for Linear MDPs only or in the more general case?

**Limitations:**

There is no limitations mentioned.

---

> ### Author Rebuttal · Authors · 2024-08-05
>
> Thank you for your critical review of our work. We understand your
> concerns regarding the linear MDP assumption and significance of our
> contribution. Due to space constraints, we directly respond to your individual questions in the "Weaknesses" and "Questions" sections according to the provided numbering. Please see below:
>
> &nbsp;
>
> **Weaknesses**
>
> **Q1.** Indeed, in the paper we focus on the linear MDP setting.
> While we agree that the linear MDP assumption is indeed restrictive
> and  may not be applicable in most real world problems, we have to
> highlight that it is one of the most widely-studied settings
> for developing efficient RL algorithms for large state
> spaces with linear function approximation, employed in hundreds of RL
> theory papers (many cited in our bibliography).
> Within this very well-studied setting, our results are the arguably
> strongest known ones, simultaneously demonstrating the best sample complexity
> and computational complexity, in the challenging infinite-horizon
> discounted-reward setting. We believe that this contribution (achieving the
> best known result in a thoroughly studied domain) is arguably strong
> enough to warrant the interest of the RL theory community. Looking forward,
> we are confident that the theoretical insights achieved in this work
> will be useful for developing RL methods
> that work under weaker assumptions such as $Q^{\pi}$-realizability and Bellman
> completeness.
>
> **Q2.** It is true that our result builds on both the work of Hong and Tewari (2024)
> and makes use of the stabilization technique of Neu and Okolo (2024) and
> Jacobsen and Cutkosky (2023). We do wish to point out that our analysis has made several
> other
> improvements to the techniques of Hong and Tewari (2024), including:
>
> 1. The removal of the cumbersome ``nearest-point oracle'' required in their
> original analysis. (Note that the latest version of their paper, online since June 2,
> has apparently also managed to remove this limitation.)
>  2. Working with a much simpler parametrization of the $\lambda$ variables.
> This allowed us to conduct a more straightforward analysis and remove some of the
> strong assumptions made in their original work (e.g., requiring very strong coverage
> assumptions - see for example the derivations in their Section D.3).
> 3. This refined analysis, together with stabilization, allowed us to also remove the
> need for prior knowledge of the coverage parameters (which are typically unavailable
> in practice).
>
> Overall, our result can be seen as the final product of a line of work
> initiated by Gabbianelli et al. (2024), refined by Hong and Tewari (2024), and
> finally perfected in the present submission.
> Arguably,
> these previous works were all suboptimal in one way or another, whereas our result
> finally demonstrates the power of this approach by achieving state-of-the art results
> across all dimensions of sample complexity, computational complexity, and
> weakness of coverage conditions. Besides these quantitative improvements, we believe
> that our analysis is much simpler and more transparent than the analyses in all of
> these previous works, and as such future researchers will have an easier time building
> further developments on top of it. We believe that these are strong enough contributions
> to warrant publication.
>
> **Q3.** As it is very common in the related literature on RL theory (incl.~just
> about all of the papers we cite), our paper focuses on theoretical performance
> guarantees. Our theoretical guarantees are indeed limited to linear MDPs, but
> nevertheless our algorithm remains well-defined for MDPs that are not linear,
> and we find it plausible that future work will be able to show guarantees for it
> under weaker assumptions. We nevertheless feel that the results in this paper are
> interesting enough as they are, and such challenging extensions have to be left
> for future work.
>
> Regarding your last statement, we are unsure what is meant by the phrase
> ``the approach is still approximate (given the bound)''. Can you please clarify?
>
> &nbsp;
>
> **Questions**
>
> **Q1.** The bound in Theorem 3.1 is good because it scales appropriately with the
> coverage parameter $\|\mathbf{\lambda}^{\*}\|\_{\Lambda_{n}^{-1}}$ which is small under the
> weakest known coverage requirement -- that $\Lambda_{n}$ sufficiently covers a
> single direction in the feature space. We refer to Appendix E. in Gabbianelli
> et al. (2024) for a detailed comparison of coverage conditions and an explanation
> as to why all other coverage conditions are more restrictive than the one we
> consider. All previous works that required such a weak assumption on data coverage
> are either achieved by impractical algorithms that are limited to finite-horizon problems (e.g., Zanette et al., 2021) or
> achieved suboptimal dependence on the precision level $\varepsilon$ (e.g.,
> Gabbianelli et al., 2024). This makes our result the strongest known one in this
> setting, which is arguably significant given how well-studied our problem is.
>
>
> **Q2.** Regarding the first part of your question, see our response above.
> Regarding the second question: Appendix E.~of Gabbianelli et al. (2024) also provides
> some answers to this. We believe that this notion of coverage is only applicable
> in linear MDPs: for such MDPs, the feature occupancy associated with a state-action
> distribution fully determines the next-state distribution. This does not necessarily
> apply to other models of linear function approximation, although some hope may be
> given by the recent results of Weisz et al. (2023) who show that $q^\pi$-realizable
> MDPs can in fact be modeled as linear MDPs. It remains to be seen if our results remain
> applicable in such more general settings.

---

### Author Rebuttal · Authors · 2024-08-05

We thank the reviewers for the detailed feedback on our work, particularly regarding the linear MDP assumption, possible application to the stochastic shortest path problem, and comparison with Zanette et al (2021). Please see our response to your individual comments and questions below.

---

### Decision · Program_Chairs · 2024-09-25

**Decision:**

Reject

**Comment:**

This paper studied offline Reinforcement Learning in large infinite-horizon discounted MDPs with linear realizability. It is based on a linear program formulation, and gradient descent in the feature occupancy space. The new algorithm enjoys favorable compuational and sample complexity guarantees, under weak data coverage assumptions. The algorithm and analysis appeared to be new to me, though the LP framework for offline RL has been well-studied in the recent literature of offline RL. There were some concerns regarding the similarity and technical novelty compared with a few literature, especially Zanette (2021) and Hong and Tewari (2024). In fact, the series of previous works by Gabbianelli et al. (2024) and Hong and Tewari (2024) diminished the novelty of the present paper. Moreover, as claimed to be a simple and "easy-to-implement" algorithm, no experimental results were implemented/compared. I agree that as a theory-oriented paper, experiments are not always necessary, but since this is an "algorithm" paper, with focus on the simplicity and implementability of the algorithm, it would have been stronger to really demonstrate this through experiments. It is a borderline paper, and I would recommend the authors incorporate the comments from this round to improve the paper, and consider resubmitting to upcoming ML venues.